# DOUBLE DESCENT REVISITED: WHEN NOISE AMPLIFIES AND OPTIMIZERS DECIDE

## ABSTRACT

We examine how the double descent phenomenon emerges across different architectures, optimisers, learning rate schedulers and noise-robust losses. Previous studies have often attributed the interpolation peak to label noise. However, by systematically varying noise levels, optimizers, learning rate regimes and training losses, we demonstrate that, while noise can amplify the effect, it is unlikely to be the driving factor behind double descent. Instead, optimization dynamics, notably learning rate and optimizers, strongly influence whether a visible peak appears, often having a larger effect than adding label noise. Consistently, noise-robust losses partially mitigate double descent in settings where the amplification effect of noise is strongest, while their impact is negligible when this is not the case. Expanding on recent work, this study further confirms that noise primarily deteriorates the linear separability of different classes in feature space. Our results reconcile seemingly conflicting prior accounts and provide practical guidance: commonly used learning rate/scheduler combinations/losses can prevent double descent, even in noisy regimes. Furthermore, our study suggests that double descent might have a lesser impact in practice. Our code is available at https://anonymous.4open.science/r/DDxNoise.

## 1 INTRODUCTION

Overparameterized neural networks have revolutionized the field of machine learning, achieving unprecedented performance in various complex tasks. However, their success has simultaneously challenged long-standing tenets of statistical learning theory, which traditionally posited that increasing model complexity beyond a certain point would inevitably lead to overfitting and poor generalization. Current large models typically have enough capacity to perfectly interpolate training data (i.e., achieve zero training error). Yet, they often generalize remarkably well to unseen data (Zhang et al., 2021), seemingly contradicting the conventional wisdom that exceedingly complex models will simply memorize training data, including noise, inevitably leading to overfitting and failing to generalize to unseen examples.

This paradox is exemplified by the Double Descent (DD) phenomenon, in which generalization error initially decreases with model size, peaks around the point of perfect training data interpolation, and then decreases again in the highly overparameterized regime (Belkin et al., 2019; Nakkiran et al., 2021a). This "second descent" implies that overparameterization, rather than being detrimental, allows models to learn more robust features or to implicitly regularize against harmful noise.

Several explanations have been offered, ranging from implicit regularization by optimization algorithms, which in the heavily overparameterized regime tend to favor solutions that generalize well (Keskar et al., 2016; Soudry et al., 2018; Ji & Telgarsky, 2019; Dar et al., 2021), to "benign overfitting" in high-dimensional settings, demonstrating that interpolating noisy data can still yield robust generalization under specific data structures (Belkin et al., 2019; Bartlett et al., 2020). Further theoretical work highlights the role of model misspecification (Mei & Montanari, 2022; Hastie et al., 2022), which leads to significant generalization error in models that are underparameterized or at the interpolation threshold, while the abundance of parameters provides the model with sufficient flexibility to effectively "compensate" in the highly overparameterized regime.

The complex interplay of these factors is essential for a comprehensive understanding of this dual risk curve. More recently, Gu et al. (2024) highlighted the crucial role that noisy samples seem to

play in shaping the behaviour of deep learning models. They show that overparameterized models interpolate more correct training data around noisy ones of matching classes, effectively isolating noise and explaining accuracy recovery after the interpolation peak.

**Our contribution.**    In this paper, we show that *DD is not caused by noisy data but is instead a phenomenon that is directly attributable to the optimization process*.

As we discuss more extensively in Section 2, past work has identified a number of potential factors behind DD, with noise often identified as a crucial one. In this study, we try to reconcile some of these different perspectives into a coherent picture. We revisit *model-wise* DD through a multi-dimensional analysis of the phenomenon, exploring the interplay between noise, optimizer and learning rate. The picture that emerges identifies some clear trends. First and foremost, noise alone, even at high levels, is not a necessary cause for DD, while it has a magnifying effect in some cases. Accordingly, we show that noise-robust losses only mildly mitigate DD by reducing the amplifying effect of noise, but offer little benefit otherwise. On the contrary, the optimization algorithm and learning rate (with and without scheduler) appear to be strong drivers of DD even in the absence of noise, something we discuss in detail in Section 4.

To better zoom in into the above phenomena, we analyze DD in feature space using k-Nearest Neighbors (kNN), expanding the analysis of Gu et al. (2024) to include various combinations of optimizers and learning rates, with or without scheduler. Moreover, differently from them, we also investigate geometry of the feature space through the lens of the Nearest Centroid (NC) heuristic. By design, this heuristic is more sensitive to misalignment between clustering of the training set in feature space and actual training set labels and as such, it has been used in the analysis of the neural collapse phenomenon (Papyan et al., 2020). Together, kNN and NC highlight complementary aspects of the geometry of learned regions. Our analysis reveals that geometry-based accuracies consistently surpass the standard linear head, particularly in the presence of noise, showing also optimizer-dependent patterns. This demonstrates that models can learn robust representations even when the final classifier struggles.

While some trends identified in this study have been separately investigated in the past, we believe our work provides a unified and fresh perspective on a very intriguing phenomenon. We also believe our work can offer cues for future research on DD, as elaborated further on in Section 5.

## 2    RELATED WORK

Investigating the double descent (DD) phenomenon—its occurrence and underlying causes—has been the focus of numerous recent studies in machine learning. A body of theoretical work examines DD in simplified contexts, often using linear models for regression (Hastie et al., 2022; Loog et al., 2020; Muthukumar et al., 2021; Gamba et al., 2022; Gu et al., 2024; Curth et al., 2023; D'Ascoli et al., 2020; Adlam & Pennington, 2020). In this work, we focus specifically on how *model-wise* DD depends on noise and on optimization process in more complex, deep models. For the sake of space, we review prior studies along these two axes, referring the reader to Appendix A for other contributions that fall outside of this scope but have some bearing on this study.

**Noise as a fundamental driver of double descent**    A substantial body of work connects the appearance of the double descent peak primarily to the presence of label noise. The seminal contributions of Belkin et al. (2019) and Nakkiran et al. (2021a) provided extensive empirical evidence across architectures and datasets, showing that the cross entropy loss curve reaches its maximum near the interpolation threshold largely because models acquire the capacity to fit noisy labels. This noise-centric perspective is strengthened by Gu et al. (2024), who showed that overparameterized models progressively "isolate" noisy samples in representation space, embedding them near clean same-class neighbors and thus reducing their detrimental effect on generalization. Similarly, Somepalli et al. (2022) analyzed decision boundaries, observing fragmentation near interpolation that stabilizes with width; they argue that noise makes DD more visible because the model must carve out local misclassified regions to fit noisy labels. A complementary bias–variance account is given by Yang et al. (2020), who showed that bias decreases monotonically with width while variance is unimodal; DD arises when the variance peak, often amplified by noise, dominates near interpolation. Studies on random feature models further confirm that variance can diverge at the threshold, with

DD observable even without noise but exacerbated by it (D'Ascoli et al., 2020; Adlam & Pennington, 2020). Together, these works establish noise as an important factor in DD, but they do not investigate other co-factors, leaving open whether noise is the key driver behind double descent or merely an amplifier.

*Noise-robust losses.* In our study, we also employ noise-robust losses as a further way to assess the impact of noise on double descent. Since these methods are not a primary research focus but practical tools in this work, we refer the interested reader to Appendix A for an overview.

**The role of optimization**   Only a handful of studies have emphasized the role of optimization dynamics and associated hyperparameters in shaping the double descent (DD) phenomenon (Kuzborskij et al., 2021; Liu & Flanigan, 2023; Nakkiran et al., 2021b; Gamba et al., 2022). Kuzborskij et al. (2021) stressed the importance of optimization, with a different focus with respect to our work. They provide a theoretical analysis of DD in least-squares regression via gradient descent, highlighting how the feature covariance spectrum drives excess risk even without noise; however, their empirical scope is limited to one-hidden-layer MLPs with square loss and shows minimal variation across learning rates. Liu & Flanigan (2023) empirically examine random feature models and two layer networks finding DD depends on feature matrix conditioning and convergence speed—slower regimes suppress peaks, while faster optimization restores them. Their work thus reinforces the findings of Kuzborskij et al. (2021), though within relatively simple models. Our work complements and extends the perspective of both works to modern deep architectures and classification tasks, providing a broader analysis of how optimizer choice, learning-rate schedules, and label noise jointly shape DD. Nakkiran et al. (2021b) also highlights the stabilizing role of regularization. They show that optimal $\ell_2$ regularization can mitigate or eliminate DD, producing monotonic risk curves in both random feature models and convolutional networks. The analysis is limited to the tuning of the regularization coefficient, without examining the influence of optimizer choice or learning-rate dynamics. Gamba et al. (2022) conducted an empirical study on loss landscape smoothness in neural networks, linking loss sharpness in input space with both model- and epoch-wise double descent.

Overall, literature converges on a multi-causal view of DD, whereby noise is a powerful amplifier and often the clearest empirical correlate for risk peak, but optimization dynamics, model misspecification, geometry of representations play decisive roles. Our study builds on this integrated perspective by disentangling the relative contributions of noise, optimizer choice, and loss functions to the appearance of DD in both model accuracy and feature space.

For the sake of space, we refer the reader to Appendix A for further, potentially interesting previous work that is less focal to this work but is related to aspects of our analysis.

## 3   METHODOLOGY

Our goal in this study was to conduct a comparative analysis across different configurations in order to empirically examine the role that certain factors play in the onset of model-wise double descent. In particular, we systematically manipulate three factors: label noise, optimizer dynamics, and training loss. We sweep model capacity across under- to over-parameterized regimes.

### 3.1   RESEARCH QUESTIONS

We organize our investigation around three guiding questions:

- **RQ1. What is the role of label noise in the manifestation of double descent? To what extent do noise-robust losses mitigate it?** Prior work (Nakkiran et al., 2021a; Somepalli et al., 2022; Yang et al., 2020) suggests that noise exacerbates risk peak near the interpolation threshold. We test whether noise is a necessary cause of DD, or merely an amplifier. To further test this, we also use noise robust losses, under the hypothesis that if DD is (mostly) noise-driven, robust losses should strongly reduce or completely eliminate it.

- **RQ2. How do optimizers, learning rates and schedulers affect double descent?** Prior studies (Liu & Flanigan, 2023; Nakkiran et al., 2021b) highlight the role of optimization

in DD emergence. We investigate whether optimizer choice and hyperparameter settings (learning rate, scheduling) can alone induce or suppress DD, even in noise-free settings.

- **RQ3. What does the geometry of learned representations reveal about the impact of DD on accuracy? How are these geometric patterns shaped by optimization and noise?** DD is usually examined through loss or accuracy curves. Building on previous representation-level analyses (Gu et al., 2024; Somepalli et al., 2022), we ask whether DD leaves distinct signatures in representation space. Unlike prior work by Gu et al. (2024), which primarily used geometry to investigate the interpolation of noisy samples, our focus is on how optimization dynamics and label noise affect local label-consistency and global cluster compactness, and how these factors explain the observed differences between standard accuracy and geometry-based accuracies.

## 3.2 Loss Functions and Noisy Labels

Most of the experiments in this study use standard Cross Entropy (CE) loss $\mathcal{L}_{\text{CE}}$ as training objective. To assess the role of label noise in DD, we also employ two loss functions designed to mitigate the impact of noisy labels: Forward Loss (Patrini et al., 2017) NCOD Loss (Wani et al., 2024).

We denote by $\mathbf{y}_i$ the one-hot encoding of $\mathbf{x}_i$'s class, i.e., so that it has $C$ components if $C$ denotes the number of classes, with the entry corresponding to $\mathbf{x}_i$'s class equal to $1$ and all other entries equal to $0$. $\hat{p}(\mathbf{y} \mid \mathbf{x}_i)$ is a vector, whose $j$-th component is the (conditional) probability assigned by the classifier to point $\mathbf{x}_i$ being of class $j$. Analogously, $\hat{\mathbf{y}}_i$ is the one-hot representation of the class predicted by the model. Finally $y_i$ (resp. $\hat{y}_i$) denote the class of the $i$-th sample (resp. the class assigned to the $i$-th sample by the classifier).

Forward correction of CE loss (FWD) (Patrini et al., 2017) is based on a label transition matrix $\mathbf{T} \in [0,1]^{C \times C}$ that models the corruption process. Each entry $T_{ij} = p(\tilde{\mathbf{y}} = \mathbf{e}_j \mid \mathbf{y} = \mathbf{e}_i) \ \forall i, j$ denotes the probability of observing a noisy label $\tilde{\mathbf{y}}$ given a clean label $\mathbf{y}$, where $\mathbf{e}_i$ is the $i$-th canonical basis vector (a one-hot vector with $1$ in position $i$ and $0$ elsewhere). Rather than comparing the prediction of the model $\hat{p}(\mathbf{y}|\mathbf{x})$ directly with the observed noisy labels, the predictive distribution is adjusted via $\mathbf{T}$, and the CE is computed against the corrected distribution: $\mathcal{L}_{FWD}(\mathbf{e}_i, \hat{p}(\mathbf{y} \mid \mathbf{x})) = -\log \sum_{j=1}^{C} T_{ji} \hat{p}(\mathbf{y} = \mathbf{e}_j \mid \mathbf{x}))$. When $\mathbf{T}$ is the identity matrix (i.e. no noise), this reduces exactly to the standard CE loss, ensuring consistency with clean-label training.

The Noisy Centroids Outlier Discounting (NCOD) loss (Wani et al., 2024), adopts a different strategy. Each class is represented by the centroid of its latent representations, and each sample $i$ is assigned soft labels $\bar{\mathbf{y}}_i$ proportional to their similarity to the class centroid. This method also introduces a trainable outlier discounting parameter $u_i$, which downweighs the contribution of examples suspected to be mislabeled. Formally, NCOD combines two objectives: $\mathcal{L}_{CE}(\hat{p}(\mathbf{y} \mid \mathbf{x}_i) + u_i \cdot \mathbf{y}_i, \bar{\mathbf{y}}_i)$, which reweighs CE with soft labels and outlier discounting, and $\frac{1}{C}\|\hat{\mathbf{y}}_i + (u_i - 1)\,\mathbf{y}_i\|_2^2$, which regulates the dynamics of $u_i$. Unlike correction-based methods, NCOD does not require prior knowledge of the noise rate or clean anchor points. Instead, it adaptively reduces the influence of outlier samples during training, making it applicable in more realistic settings. This discourages overfitting mislabeled samples in later epochs, while preserving the signal from clean ones. We adopt NCOD because it builds on SOP (Liu et al., 2022) but achieves better empirical performance in noisy environments.

## 3.3 Evaluation and Geometry-based Metrics

We evaluate models with three complementary defined metrics that capture (i) probabilistic/generalization quality, (ii) performance of the trained linear head, and (iii) geometric structure of the learned representation.

As a loss-based metric, we report the average CE on the test set since DD is most commonly observed in loss curves (Belkin et al., 2019; Nakkiran et al., 2021a). We use CE as a uniform evaluation metric across all experiments, even when training with FWD or NCOD. Since these methods only modify the training loss, all test results are computed with CE. This ensures comparability across settings and isolates the effect of the training dynamics.

Soft-label accuracy measures the standard classification performance obtained from the model's final linear layer. In this section, we define $\hat{y}_i = \arg\max_c \hat{p}(\mathbf{y}_i \mid \mathbf{x})$ the class predicted by the model for the $i$-th sample in the dataset.

Accuracy is computed as $\mathcal{A}_{\text{acc}} = \mathcal{A}(\{\hat{y}_i\}_{i=1}^M)$ with $\mathcal{A}(\{\hat{y}_i\}_{i=1}^M) = \frac{1}{M}\sum_{i=1}^M \mathbf{1}\{\hat{y}_i = y_i\}$. Conceptually, this metric evaluates the separating hyperplane learned by the final linear head in the penultimate-layer feature space. Therefore, it is sensitive to both feature geometry and how the classifier converts features into calibrated probabilities.

Standard evaluation metrics such as CE and accuracy capture prediction quality but provide limited insight into the geometry of feature space, especially when they are misaligned. In practice, cross-entropy may be high, indicating low confidence, while accuracy remains stable, suggesting that predictions still fall within correct decision boundaries. To better assess this structure, we employ two test-time decision rules that operate directly on penultimate-layer features.

**k-Nearest Neighbours (kNN) Accuracy** For each test point representation in the penultimate layer latent space $\mathbf{z}_i^{\text{test}}$, we first find its $k$ nearest training representations $\mathcal{N}_k(\mathbf{z}_i^{\text{test}})$ according to cosine distance. We then assign a label to $\mathbf{z}_i^{\text{test}}$ by majority vote: $\hat{y}_i^{\text{kNN}} = \arg\max_{c\in\{1,\ldots,C\}} \sum_{j\in\mathcal{N}_k(\mathbf{z}_i^{\text{test}})} \mathbf{1}\{y_j^{\text{train}} = c\}$. k-NN Accuracy $\mathcal{A}_{\text{kNN}} = \mathcal{A}(\{\hat{y}_i^{\text{kNN}}\}_{i=1}^M)$ measures local label-consistency in representation space: high kNN accuracy indicates that nearby training points tend to share labels regardless of the learned linear head.

**Nearest Centroid (NC) Accuracy** For each class $c$, we compute the centroid $\mu_c$ of its training latent representations $\mathcal{Z}_c = \{\mathbf{z}_i^{\text{train}}$ s.t. $y_i^{\text{train}} = c \ \forall i \in \{1,\ldots,M^{\text{train}}\}\}$. A test point $\mathbf{z}^{\text{test}}$ is assigned the label of the closest centroid under cosine distance $d$: $\hat{y}_i^{\text{NC}} = \arg\min_{c\in\{1,\ldots,C\}} d(\mathbf{z}_i^{\text{test}}, \mu_c)$.

NC accuracy $\mathcal{A}_{\text{NC}} = \mathcal{A}(\{\hat{y}_i^{\text{NC}}\}_{i=1}^M)$ quantifies global cluster compactness and class separability.

### 3.4 EXPERIMENTAL SETUP

We now describe the datasets, architectures, optimizers, and training protocol used in our study. Our goal is to ensure that comparisons across conditions are fair and reproducible, and that all factors except the one under investigation are held constant. Full implementation details are provided in Appendix B.

**Datasets.** Following Gu et al. (2024), we use two standard image classification benchmarks: CIFAR-10 (Krizhevsky et al. (2014)) and MNIST (Lecun et al. (1998)). For CIFAR-10, we use 40k/10k training/validation split, as well as the standard 10k test set. For MNIST, we use a reduced training budget of 4k/1k training/validation examples, and the standard test split. Label noise is injected into the training set by randomly replacing labels: for a given noise rate $\eta$, each training label is independently replaced with a random class label (uniformly distributed across the other $C-1$ classes). We report results for a noise-free regime ($\eta = 0$) and several noisy regimes ($\eta \in \{0.1, 0.2, 0.4, 0.8\}$).

**Architectures.** We study three model families used in prior DD work: (i) fully connected networks (FCN), (ii) small convolutional neural networks (CNN), and (iii) ResNet-18 variants. These families and configurations are drawn from Gu et al. (2024) and Nakkiran et al. (2021a). We systematically vary base layer width to span the under-to over-parameterized regimes.

**Optimization** We compared three widely used optimization strategies: vanilla Stochastic Gradient Descent (SGD), SGD with the learning rate scheduler (SGD+CS) used in Gu et al. (2024), and Adaptive Moment Estimation (Adam, Kingma & Ba (2015)). To probe sensitivity to learning rate scale, we use a log-spaced grid of five learning rates, $\{10^{-x}, x \in \{1, 2, 3, 4, 5\}\}$, plus the one used in Gu et al. (2024), 0.05.

**Training Protocol.** All models are trained for a fixed number of epochs (4,000 for FCN, 200 for CNN and ResNet) without early stopping. The training loss reaches zero within the specified number

of epochs in all cases, and corresponding training curves are reported in Appendix E for reference. Reported metrics use the final-epoch checkpoint to ensure consistent "last model" evaluations across experiments. Experiments were implemented in PyTorch [1].

# 4 EXPERIMENTAL ANALYSIS

Our experiments provide a nuanced picture of how DD arises and under which conditions it is amplified or mitigated. This section presents the results for the ResNet-18 variants; the corresponding results for FCN and CNN variants are presented in Appendix C.

## 4.1 NOISE AMPLIFIES BUT DOES NOT DETERMINE DD (RQ1)

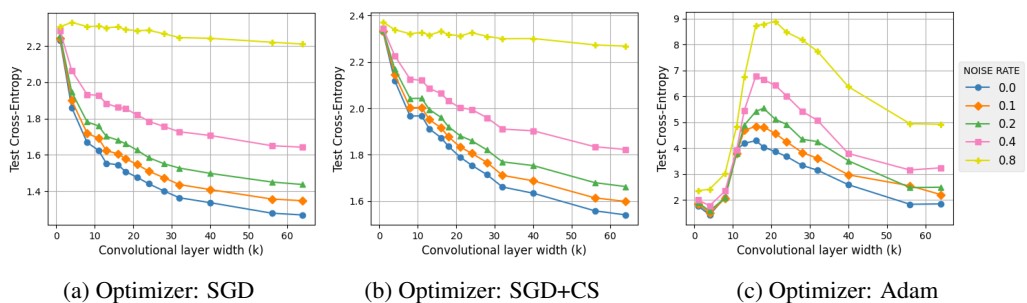

(a) Optimizer: SGD                (b) Optimizer: SGD+CS                (c) Optimizer: Adam

Figure 1: Cross Entropy on the test set as a function of the base convolutional layer width (k) of ResNet models trained with Cross Entropy loss on CIFAR-10 using three optimizers with initial learning rate 0.0001. Results show how different optimizers and noise rates affect the test error curve.

We address RQ1 by increasing model width while varying (i) label noise rate, (ii) optimizer family (SGD, SGD with scheduler, Adam), and (iii) training loss (Cross-Entropy (CE), Cross-Entropy with forward correction (FWD), and NCOD).

Fig. 1 shows clear patterns. Using Adam (Fig. 1c), we observe a significant double-descent peak, consistent with Gu et al. (2024); importantly, this peak is visible even when noise rate is zero. By contrast, the SGD baseline (Fig. 1a) and SGD combined with a scheduler (Fig. 1b) do not display a comparable peak, even when label noise is extreme (80% on a 10-class task). The results obtained for ResNet architectures exhibit patterns consistent with those observed for CNN and FCN variants, as presented in Figs. 5 and 6. These comparisons show that label noise amplifies the magnitude of the peak, but it is not a necessary condition for its appearance: optimization dynamics can alone produce or suppress DD.

To isolate the effect of noise-robust losses, Fig. 2 compares CE and accuracy curves for CE, FWD, and NCOD at three noise levels (0.0, 0.1, 0.2) using the Adam optimizer. Three consistent observations emerge. First, the amplitude of the CE peak scales with the noise rate: higher noise rate increases the peak and makes the associated accuracy dip more pronounced. Secondly, the CE peak is not eliminated by robust loss correction, but only marginally mitigated. Third, under moderate to high noise, robust losses substantially reduce the "*double-ascent*" in accuracy for small and intermediate widths (with NCOD showing the largest reduction). However, the benefit of robust losses largely tends to vanish when width becomes large, as CE, FWD and NCOD converge to similar accuracies. In other words, robust losses mitigate the *accuracy* degradation induced by noise in underparameterized or mid-scale regimes, but they do not eliminate the CE peak across all regimes.

Although label noise amplifies the loss peak and the associated accuracy dip, it is neither the sole nor the necessary cause of double descent. Optimization process, particularly the optimizer family, play a primary role in determining whether a visible peak appears, while noise mainly acts as a multiplier of that effect. Although noise-robust losses are effective in recovering accuracy in small

---

[1] https://anonymous.4open.science/r/DDxNoise

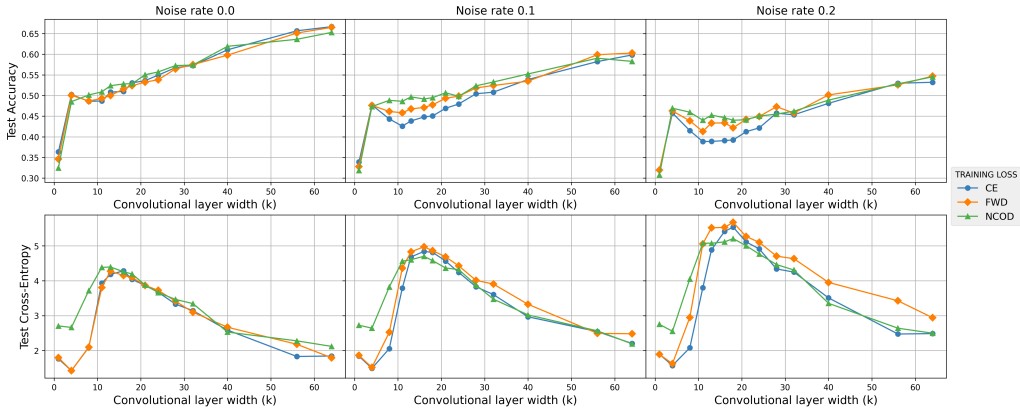

Figure 2: Accuracy (top row) and Cross-Entropy (bottom row) on the test set as functions of the base convolutional layer width (k) of ResNet models trained on CIFAR-10 using Adam optimizer with an initial learning rate of 0.0001. Results are reported for noise rates (0.0, 0.1, 0.2) under varying training losses: Cross-Entropy (CE), Forward-corrected CE (FWD) and NCOD.

and intermediate models where the amplifying effect of noise is strongest, they do not universally eliminate the CE peak in large models.

## 4.2 ROLE OF OPTIMIZER AND LEARNING RATE (RQ2)

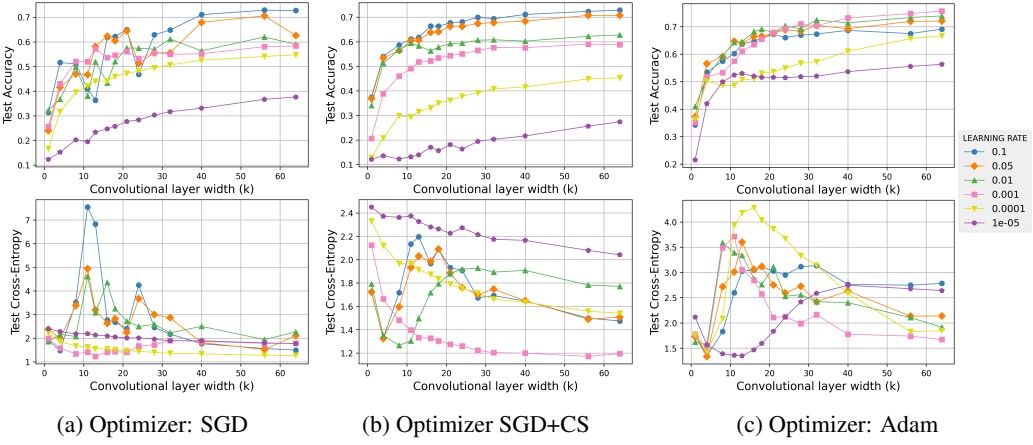

(a) Optimizer: SGD      (b) Optimizer SGD+CS      (c) Optimizer: Adam

Figure 3: Accuracy (top row) and Cross-Entropy (bottom row) on the test set as functions of the base convolutional layer width (k) of ResNet models trained on CIFAR-10 using the three optimizers and multiple initial learning rates. Noise rate is 0.0.

Fig. 3 shows the test accuracy and CE for three optimizers (SGD, SGD with scheduler and Adam), each of which was trained using six different learning rates, with no added noise. Learning rate has a significant impact on the appearance of DD and the amplitude of its peak for Vanilla SGD (Fig. 3a),. At relatively high learning rates, a visible double-descent peak in CE test-loss is consistently observed. Reducing learning rate lowers the peak's amplitude and, in some cases, eliminates it altogether. This suggests that learning rate modulates the optimization dynamics: higher rates induce less stable trajectories, thereby revealing the double-descent phenomenon.

Adoption of a learning rate scheduler (Fig. 3b), as in Gu et al. (2024) does not change this picture: larger initial learning rates amplify the peak, while smaller rates decrease or remove it. Moreover, while CE still exhibits peaks in both cases, test accuracy curves do not display significant double ascent phenomena. In particular, intermediate-sized models exhibit a relatively constant upward trend as width increases rather than sudden drops, suggesting that the scheduler stabilizes training

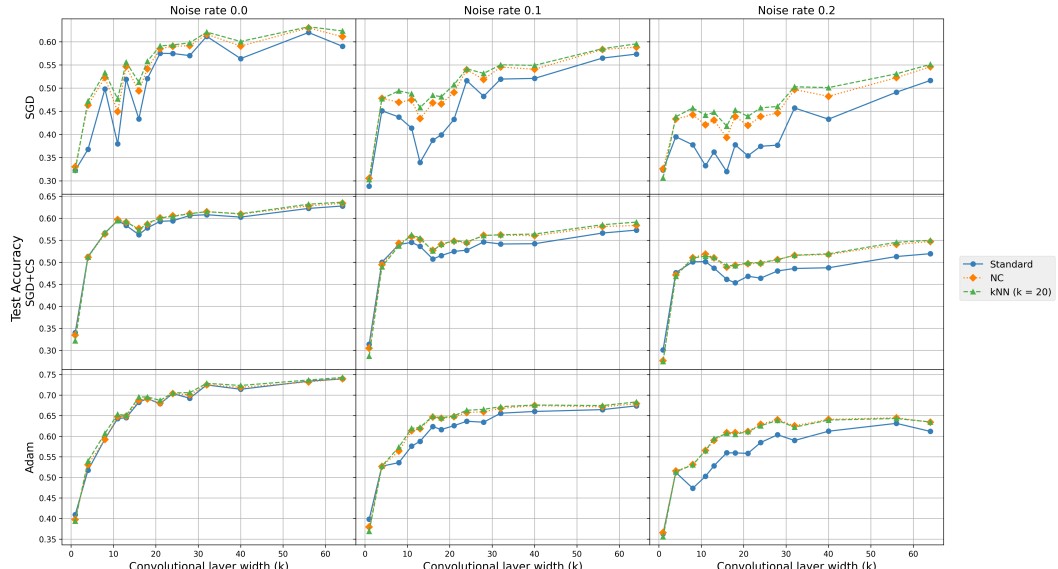

Figure 4: Test accuracy on the test set as function of the base convolutional layer width (k) of ResNet models trained on CIFAR-10 with Cross-Entropy loss. The plot compares model classifiers predictions (Standard) with Nearest Centroid (NC) and k-Nearest Neighbours (kNN, k=20) predictions under varying label noise rates (0.0, 0.1, 0.2) and optimizers (SGD, SGD+CS, Adam) with initial learning rate 0.01.

sufficiently to lead to smoother convergence. This results in higher learning rates always leading to better performance in terms of accuracy.

Switching to Adam (Fig. 3c) radically changes the picture, even keeping the very same experimental configuration. In this case, double descent persists across all tested learning rates. While amplitude, shape and location of the peak vary with the learning rate, the phenomenon itself never disappears. This suggests that Adam's internal adaptation mechanisms (e.g. momentum terms) introduce dynamics that are less sensitive to global learning-rate scaling. Moreover, unlike SGD (with or without a scheduler), the best accuracy is not achieved for the largest learning rate; rather, accuracy tends to plateau at intermediate values.

Similar patterns are observed across CNN and FCN variants (Figs. 9 and 10), confirming the generality of these optimizer-dependent dynamics. Taken together, these results support the hypothesis that optimization dynamics, both optimizer and learning rate, fundamentally determine whether double descent arises and how strongly it affects performance, even in the absence of noise.

## 4.3 REPRESENTATION-SPACE EVALUATION (RQ3)

In order to more closely look into the reasons underpinning the frequent misalignment between test entropy loss and accuracy, we repeat previous experiments, this time probing the geometry of the learned feature space through kNN and NC accuracies. These metrics test whether the representation organizes samples into semantically meaningful local neighborhoods (kNN) and/or clusters (NC), regardless of the separating hyperplane learned by the final layer.

Fig. 4 reports results across optimizers (rows) and noise rates (columns), with (initial) learning rate 0.01, a value that consistently corresponds to well-defined DD peaks in CE loss across all classifiers (see Fig. 3). Across conditions, kNN and NC accuracies exceed the standard accuracy of the final classifier. This suggests that networks learn a latent space with a strong geometric structure, where samples from the same class cluster together and/or share local neighborhoods, even when the final, linear classifier cannot find a clear separating hyperplane. This effect intensifies at higher noise rates, as the difference between standard and geometry-based accuracy widens. This suggests that noise makes linear separation more fragile, while the underlying representation remains consistent, with feature-space aggregation providing implicit denoising.

Something interesting appears when we compare optimizers: SGD with a scheduler and Adam produce similar results for kNN and NC heuristics, with negligible double ascent in accuracy. In contrast, kNN outperforms NC when the SGD baseline is used, an effect that is amplified by noise. This suggests that learned representations tend to form irregular manifolds where local neighborhoods are label-consistent, but global centroids are less representative. This observation aligns with our earlier observation that SGD without scheduling produces less stable convergence. Consistently with these findings, accuracy of geometry-based heuristics outperforms standard accuracy in CNN and FCN architectures (Figs. 11 and 12), confirming that models can learn effective representations before feature collapse.

While (larger) models achieve zero training error (see Appendix E), the systematic discrepancy between the three accuracies indicates that neural collapse -the tight convergence of feature vectors to class centroids (Nguyen et al. (2023))- has not occurred in our regimes, for otherwise the three metrics would themselves collapse. Though neural collapse is not a focus of this work, we explicitly verified this condition, by tracking neural collapse metrics during training (see Fig. 13 in Appendix D). We remark that the absence of collapse is not a shortcoming; rather, it reflects realistic training regimes in which overparameterized models are trained long enough to interpolate training data, but not enough for class representations to collapse into an equiangular simplex. This non-collapse regime is precisely where geometric probes are most informative; they reveal useful structure in representations before and independently of any eventual collapse.

## 5 Discussion, limitations, and conclusions

Our findings suggest that the optimization process is the main driver of the double descent (DD) phenomenon. Label noise, while not essential for its appearance, can significantly amplify it, as confirmed by the behavior of the phenomenon in the presence of noise-robust losses. Our analysis reveals that the choice of optimizer, learning rate, and possibly scheduling are dominant factors; for instance, learning rate can either induce or eliminate DD with SGD, whereas Adam consistently exhibits it. Representation probes show that even when soft-label accuracy temporarily declines, overparameterized models organize features into geometrically robust manifolds. This explains why k-NN and Nearest Centroid (NC) accuracies can still outperform standard accuracy in these cases.

While these results shed light on key mechanisms underpinning DD, their practical impact must be qualified. As is common in much of the DD literature, our performance curves were obtained by evaluating the final model at the end of the last training epoch. However, in realistic training pipelines, models are typically selected using a validation set with early stopping or best-checkpoint selection. When this standard procedure is applied, the characteristic DD peak often disappears, as the validation set apparently prevents the model from reaching the stages where DD would manifest (see Appendix F and Fig. 21 and Nakkiran et al. (2021a) for results of a similar flavor using early stopping). This suggests that while DD is theoretically important for understanding overparameterized models, its impact in practice is limited (Jeffares & van der Schaar (2025)).

Our analysis focused exclusively on model-wise double descent, intentionally not considering data-wise or epoch-wise variants. While this choice enabled a thorough investigation into the roles of noise, optimizers, and losses, it limits the scope of our conclusions. Similarly, while we compared cross-entropy with its noise-robust variants, we did not explore a broader range of loss functions, such as contrastive or margin-based formulations, nor did we explore the role of hyperparameters such as batch size. Finally, we mostly mentioned neural collapse to remark that our models are relatively far from achieving it at the end of training. On the contrary, a systematic study of its relationships to DD could be an interesting extension, albeit one probably deserving a separate study.

While these limits clearly identify possible extensions, an intriguing direction we did not consider concerns the statistical nature of double descent itself. Recent results show that DD can be understood within the classical bias-variance framework for simpler models Curth et al. (2023), which begs the question of whether similar arguments extend to modern deep neural networks. At the same time, our results suggest that theoretical findings involving gradient-based baselines and/or simpler models might not straightforwardly carry over to deep models.

## REPRODUCIBILITY STATEMENT

All implementation details and hyperparameter settings required to reproduce our results are provided in Section 3.4 and in Appendix B. The source code is available at our GitHub repository `https://anonymous.4open.science/r/DDxNoise`.

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

# A RELATED WORK

## A.1 EMBEDDINGS/REPRESENTATION-SPACE

From a statistical learning perspective, analyses of ridgeless least squares and random features (Hastie et al., 2022; Loog et al., 2020) attribute the second descent to implicit biases of high-dimensional solutions, while Muthukumar et al. (2021) show that classification losses exacerbate non-monotonicity compared to regression. Other explanations take a geometric lens, linking DD to the learning of smoother interpolants in overparameterized networks (Gamba et al., 2022) or to the discovery of structured, "prunable" solutions (Chang et al., 2021; Teague, 2022). Representation-space analyses, such as Gu et al. (2024), emphasize how overparameterized models reorganize noisy samples, while Curth et al. (2023) argue that DD can vanish when complexity is measured via effective degrees of freedom rather than raw parameter counts, though their results are limited to simple, non-deep models. Finally, some works suggest that DD may partly reflect experimental artifacts: Chaudhary et al. (2023) argue that the peak reported by Belkin et al. (2020) disappears once asymmetric learning-rate schedules and weight reuse are controlled.

## A.2 TRAINING WITH NOISE-ROBUST LOSSES

A central challenge in training deep neural networks is their tendency to overfit when labels are noisy, and a variety of approaches have been proposed to design loss functions that are more robust in such settings. One early contribution in this direction is by Patrini et al. (2017), who developed a practical method for correcting the loss function itself, showing experimentally that this correction can substantially improve performance under label noise. Building on the idea of modifying the loss, Zhang & Sabuncu (2018) introduced the generalized cross-entropy loss, which interpolates between mean absolute error and cross-entropy, and demonstrated that it achieves significantly greater robustness to noisy labels compared to standard cross-entropy.

Beyond proposing new loss formulations, researchers have also explored the role of model capacity in handling noise. Liu et al. (2022) provided both theoretical and experimental evidence that over-parameterization can actually help models withstand label corruption when combined with a specialized loss. Their Sparse Over-Parametrisation (SOP) method is motivated by the observation that label noise is typically sparse, and therefore can be modeled explicitly. In SOP, each noisy label is expressed as the sum of the network's prediction $f(x_i; \theta)$ and a noise term $s_i$, with the latter parameterized as $s_i = u_i \odot u_i - v_i \odot v_i$ to enforce sparsity. Both the model parameters $\theta$ and the noise parameters $u_i, v_i$ are optimized jointly through gradient descent, with careful initialization and learning-rate strategies ensuring that the noise component remains well-regularized.

More recently, Wani et al. (2024) introduced NCOD, which incorporates noise modeling directly into the latent representation space. In their framework, each class is associated with a centroid in the embedding space, computed as the average of its samples, and every input is compared against this centroid to measure similarity. At the same time, each sample is assigned a learnable weight $u_i \in [0, 1]$ that governs its contribution to the loss function. Samples that are persistently misclassified by the model—likely to be mislabeled—gradually receive larger $u_i$ values, which in turn downweight their influence during training. This mechanism enables the model to focus on clean data while automatically reducing the impact of suspected noisy labels.

## A.3 NEURAL COLLAPSE

Neural Collapse is a phenomenon observed in overparameterized neural networks during the terminal phase of training, after the training error reaches zero Papyan et al. (2020). It is characterized by highly symmetric last-layer features and classifier weights that converge toward a Simplex Equiangular Tight Frame. Papyan et al. demonstrated its prevalence across multiple architectures -ResNet, VGG, DenseNet- and datasets, and suggested that this structure can improve robustness, interpretability, and potentially generalization. Subsequent works have expanded the empirical and theoretical understanding of neural collapse: for instance, Han et al. (2021) observed neural collapse under MSE loss, while Mixon et al. (2022) showed how gradient descent dynamics drive convergence toward NC. Zhu et al. (2021) further analyzed the optimization landscape, concluding that each critical point is either a global minimum corresponding to the Simplex ETF or a saddle point with negative curvature, and therefore the neural collapse can be achieved by any optimization

method able to avoid inflection points. Other studies, such as Hui et al. (2022), emphasize that larger collapse during training does not always lead to better performance in terms of generalization, introducing the distinction between neural collapse on train set and on test set. Conversely, Galanti et al. (2021) report that neural collapse can support feature transfer and generalization to new classes, explaining its potential relevance for few-shot and foundation models.

## B  EXPERIMENTAL SETUP

All experiments were implemented in PyTorch. The full codebase is available in our GitHub repository at the following link: `https://anonymous.4open.science/r/DDxNoise/README.md`.

### B.1  MODELS

All models are based on those used by Gu et al. (2024), which in turn were based on the CNN and ResNet architectures proposed by Nakkiran et al. (2021a).

- **Fully Connected Network (FCNN):** the model consists of a single hidden layer of variable width $k$, with $k \in \{1, 3, 4, 7, 10, 13, 15, 25, 40, 100, 250, 500, 1000\}$, followed by a linear classifier. Differently from Gu et al. (2024), who utilize custom weight initialization schemes (Xavier uniform for $k = 1$, a normal distribution with parameters $(0, 0.1)$ for $k > 1$), we adopted the He initialization (He et al., 2015) (i.e. a uniform distribution between $(-b, b)$, with $b = \sqrt{3/d}$ and $d$ the number of features of the dataset) for all values of $k$.

- **Parametric CNN:** a 5-layer architecture, with four convolutional layers doubling the number of channels progressively ($[k, 2k, 4k, 8k]$, for $k \in \{1, 2, 4, 7, 10, 13, 17, 20, 25, 30, 40, 50, 64\}$), each followed by a max pooling layer with kernel sizes $[2, 2, 2, 4]$ respectively. All convolutional layers use kernel size 3, stride 1, and padding 1. A final linear classifier maps features to output logits.

- **Parametric ResNet18**: a network with four residual blocks, each containing two batch-normalized convolutional layers. Each convolutional layer progressively scales the channel dimension as $[k, 2k, 4k, 8k]$, with $k \in \{1, 4, 8, 11, 13, 16, 18, 21, 24, 28, 32, 40, 56, 64\}$. The layers operate with kernel size 3, stride $[1, 2, 2, 2]$ for downsampling, and skip connections characteristic of the ResNet architecture. A global average pooling precedes the final linear classifier. Setting $k = 64$ recovers the standard ResNet18.

### B.2  OPTIMIZATION

We trained the models using three different optimizers, widely used in the literature: Stochastic Gradient Descent (SGD), Stochastic Gradient Descent with custom learning rate scheduler (SGD + CS) and Adam.

The custom scheduler is based on the one introduced by Gu et al. (2024): $\mathrm{lr} = \frac{\mathrm{lr}_0}{\sqrt{1 + [\mathrm{epoch}/50]}}$ for FCNNs and $\mathrm{lr} = \frac{\mathrm{lr}_0}{\sqrt{1 + [\mathrm{epoch}*10]}}$ for CNN and ResNet, where $\mathrm{lr}_0$ is the initial learning rate. In both cases, the learning rate is updated every 50 epochs. FCNNs were trained for 4000 epochs, while CNNs and ResNets were trained for 200 epochs. These epochs guarantee that zero training error is achieved, as can be seen in Appendix E.

Apart from learning rate, the default PyTorch hyperparameters were kept for the optimizers: momentum = 0 for SGD, betas = (0.9, 0.999) for Adam, and weight decay = 0 for both.

Initial learning rate are chosen from $\{10^{-5}, 10^{-4}, 10^{-3}, 10^{-2}, 0.05, 10^{-1}\}$. The batch size was set to 512 across all runs. No data augmentation or regularization techniques were used.

### B.3  HARDWARE

All experiments were conducted on a Linux machine equipped with an AMD Ryzen 9 7950X 16-Core Processor, 128 GB RAM, and 2 × NVIDIA GeForce RTX 4090 GPUs. The operating system was Ubuntu 24.04.2 LTS with kernel version 6.14.0-27-generic.

# C   EXPERIMENTAL ANALYSIS ON FCN AND CNN ARCHITECTURES

This section reports the experimental results obtained with the FCN and CNN models. Overall, the behaviour observed in these two architectures is consistent with the findings reported in the main body of the paper and show that the phenomena discussed in the main text are not specific to ResNet models. Although the quantitative details vary, the qualitative patterns are remarkably aligned, strengthening the robustness of our conclusions.

## C.1   EFFECT OF LABEL NOISE.

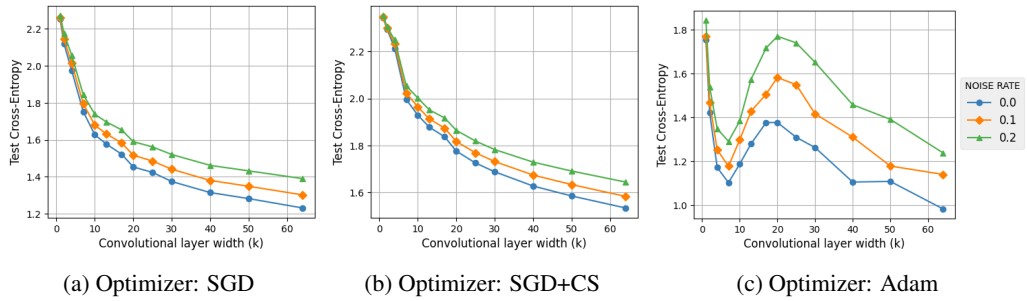

|           |           |           |
|-----------|-----------|-----------|
| (a) Optimizer: SGD | (b) Optimizer: SGD+CS | (c) Optimizer: Adam |

Figure 5: Cross Entropy on the test set as a function of the base convolutional layer width (k) of CNN models trained with Cross-Entropy loss on CIFAR-10 using three optimizers with initial learning rate 0.0001. Results show how different optimizers and noise rates affect the test error curve.

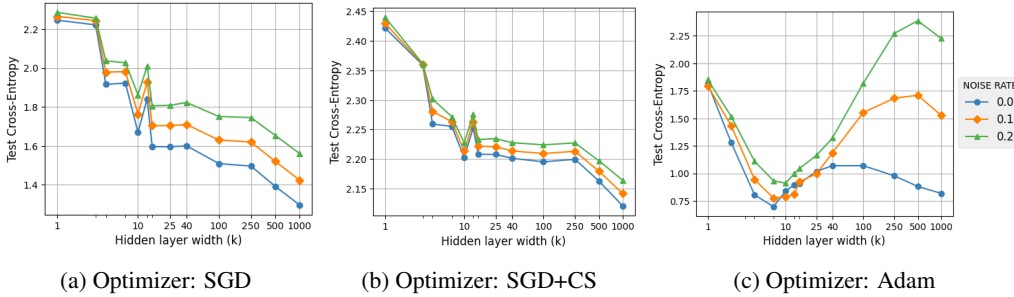

|           |           |           |
|-----------|-----------|-----------|
| (a) Optimizer: SGD | (b) Optimizer: SGD+CS | (c) Optimizer: Adam |

Figure 6: Cross Entropy on the test set as a function of the base hidden layer width (k) of FCN models trained with Cross-Entropy loss on MNIST using three optimizers with initial learning rate 0.0001. Results show how different optimizers and noise rates affect the test error curve.

Figs. 5 and 6 illustrate CE curves on the test set for CNN and FC networks under different label noise rates. In line with ResNet results, we observe that increasing the noise rate generally leads to an enhancement in the hight and visibility of the DD peak. However, noise is not the primary trigger, as DD may not emerge in presence of noise. This finding reinforces the paper's primary message: noise acts primarily as a amplifier, rather than a necessary cause of DD.

The impact of noise-robust losses is instead slightly architecture-dependent. For CNNs, as illustrated in Fig. 7, robust losses only moderately mitigate the DD peak but substantially reduce the double-ascent in test accuracy at intermediate widths. For FCNs, as depicted inFig. 8, the losses considerably improve the accuracy, reducing the accuracy dip obtained when training with CE loss, and are also slightly more effective in attenuating the CE peak. This finding suggests that simpler, fully connected architectures are more responsive to robust loss functions, although the overall qualitative behavior mirrors that of the other networks.

## C.2   EFFECT OF LEARNING RATE

Figs. 9 and 10 illustrate the impact of learning rate on DD under different optimizers. For both CNNs and FCs trained with SGD and SGD+CS, higher (initial) learning rates amplify the DD peak,

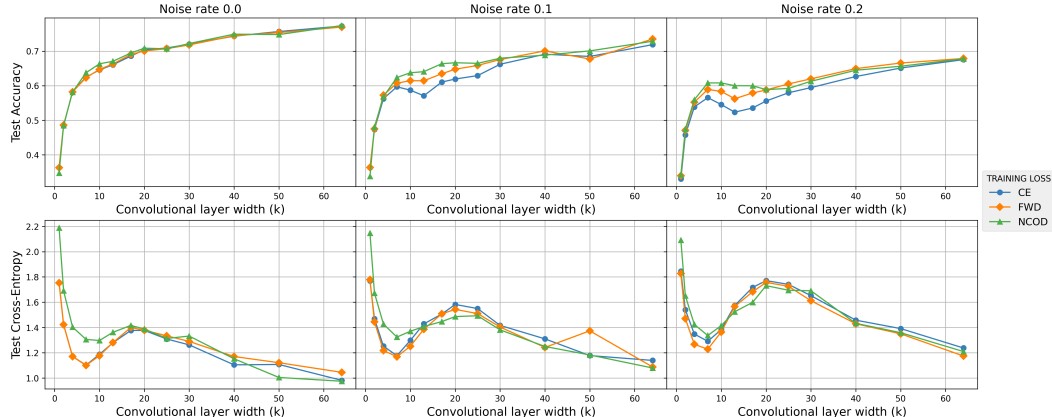

Figure 7: Accuracy (top row) and Cross-Entropy (bottom row) on the test set as functions of the base convolutional layer width (k) of CNN models trained on CIFAR-10 using Adam optimizer with an initial learning rate of 0.0001. Results are reported for noise rates (0.0, 0.1, 0.2) under varying training losses: Cross-Entropy (CE), Forward-corrected CE (FWD) and NCOD.

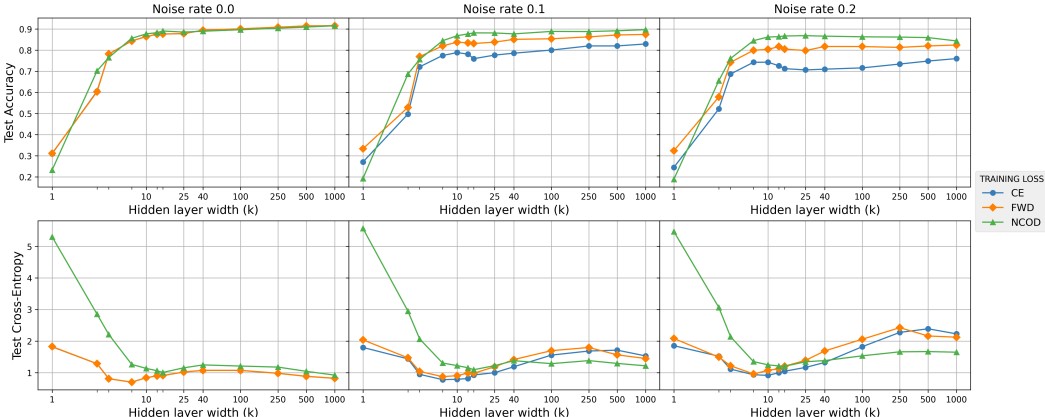

Figure 8: Accuracy (top row) and Cross-Entropy (bottom row) on the test set as functions of the base hidden layer width (k) of FCN models trained on MNIST using Adam optimizer with an initial learning rate of 0.0001. Results are reported for noise rates (0.0, 0.1, 0.2) under varying training losses: Cross-Entropy (CE), Forward-corrected CE (FWD) and NCOD.

while lower rates tend to attenuate or suppress it entirely. As observed with ResNet, Adam preserves DD across all tested learning rates, thereby further underscoring the notion that optimizer choice can dominate the effects of learning rate in determining the appearance and magnitude of DD. A practical note: high learning ($lr = 0.1$) and noise (0.2) rates produce very high loss values for FCNs optimized with Adam; for clarity, these plots use a logarithmic $y$-axis.

## C.3 ACCURACY METRICS.

We compare standard softmax accuracy with geometry-based metrics, kNN and NC accuracy, across both architectures in Figs. 11 and 12. In most cases, the three metrics do not perfectly align, but the trend we observed in Section 4.3 for ResNet is still visible: geometry-based accuracies generally outperform the standard softmax measure. The analysis of neural collapse metrics (Figs. 14 and 15) confirms that even with this architectures, full collapse is not achieved. That is, while the models interpolate the training data, their features have not yet converged to class centroids, leaving room for discrepancies between standard and geometry-based accuracies. This further supports the interpretation that overparameterized networks can develop good latent representations, even in the presence of noise, well before achieving complete neural collapse.

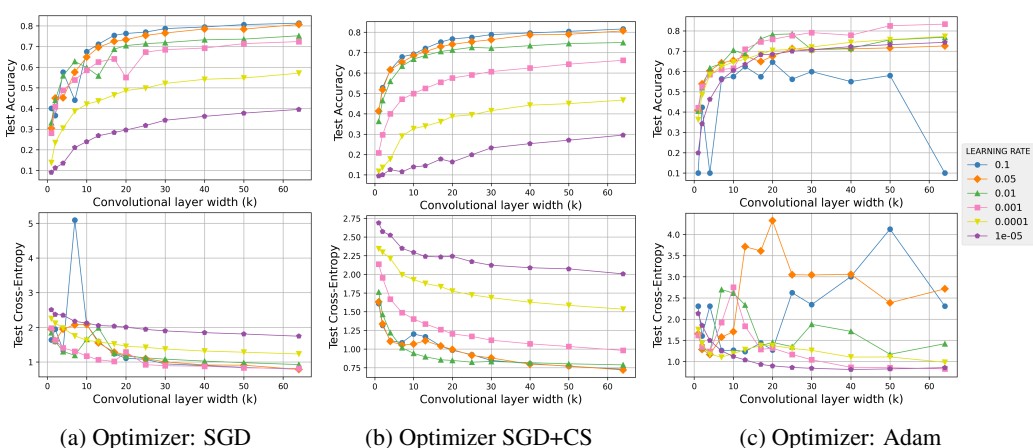

(a) Optimizer: SGD        (b) Optimizer SGD+CS        (c) Optimizer: Adam

Figure 9: Accuracy (top row) and Cross-Entropy (bottom row) on the test set as functions of the base convolutional layer width (k) of CNN models trained on CIFAR-10 with Cross-Entropy loss, using the three optimizers and multiple initial learning rates.

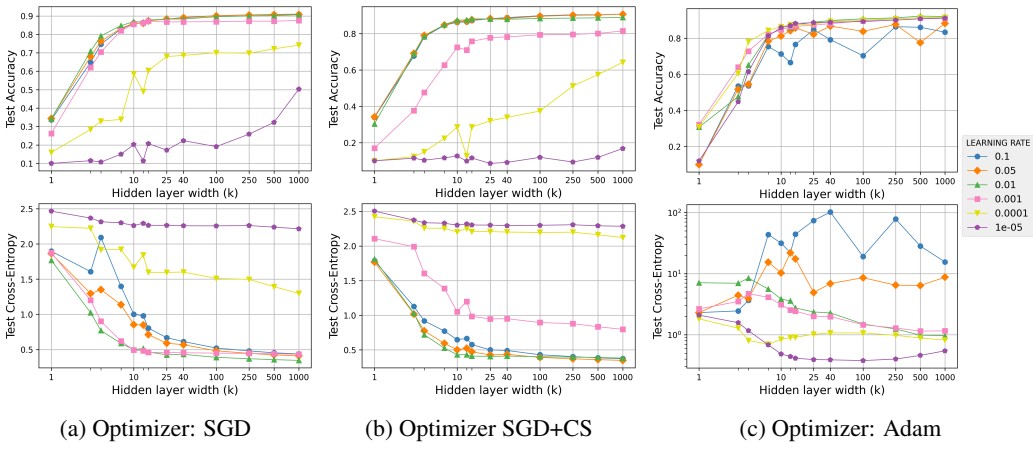

(a) Optimizer: SGD        (b) Optimizer SGD+CS        (c) Optimizer: Adam

Figure 10: Accuracy (top row) and Cross-Entropy (bottom row) on the test set as functions of the base hidden layer width (k) of FCN models trained on MNIST with Cross-Entropy loss using the three optimizers and multiple initial learning rates. The last plot is shown on a logarithmic scale, since the Cross-Entropy with learning rate 0.1 reached very large values.

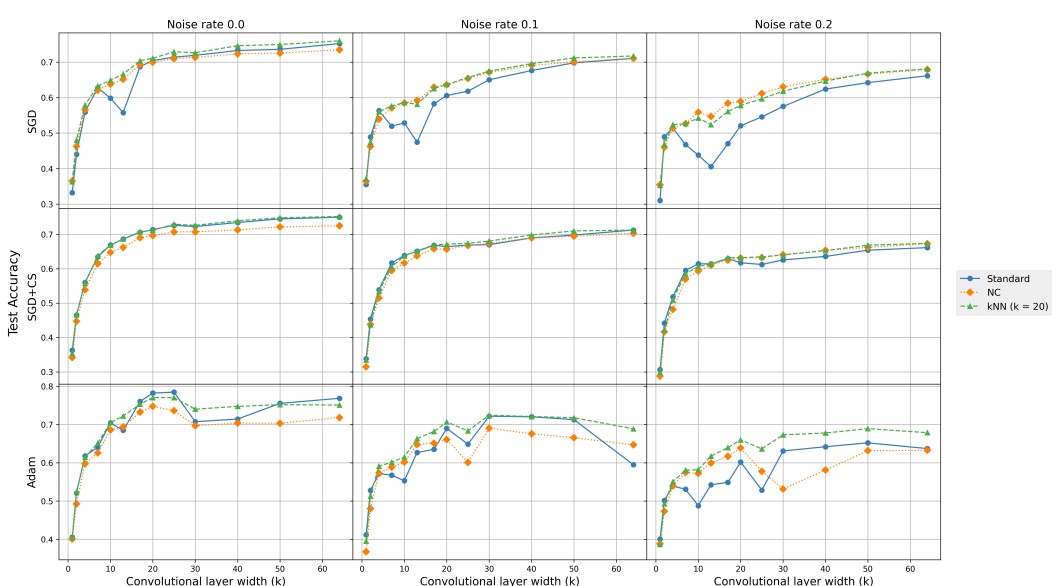

Figure 11: Test accuracy on the test set as function of the base convolutional layer width (k) of CNN models trained on CIFAR-10 with Cross-Entropy loss. The plot compares model classifier predictions (Standard) with Nearest Centroid (NC) and k-Nearest Neighbours (kNN, k=20) predictions under varying label noise rates (0.0, 0.1, 0.2) and optimizers (SGD, SGD+CS, Adam) with initial learning rate 0.01.

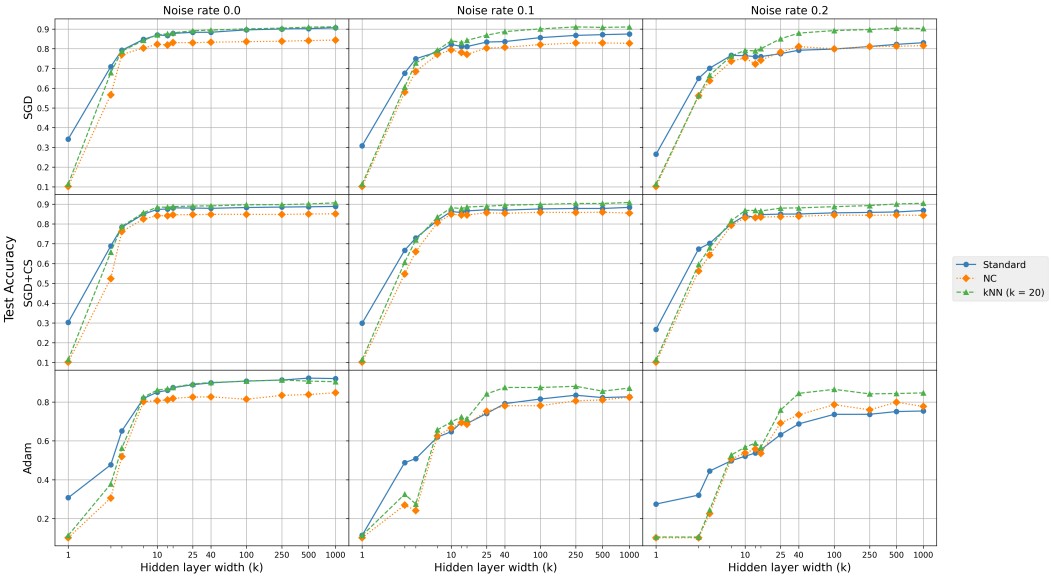

Figure 12: Test accuracy on the test set as function of the base hidden layer width (k) of FCN models trained on MNIST with Cross-Entropy loss. The plot compares model classifiers predictions (Standard) with Nearest Centroid (NC) and k-Nearest Neighbours (kNN, k=20) predictions under varying label noise rates (0.0, 0.1, 0.2) and optimizers (SGD, SGD+CS, Adam) with initial learning rate 0.01.

Taken together, these findings confirm that the key phenomena observed in ResNet models generalize to FCN and CNN architectures: (i) label noise amplifies but does not cause DD; (ii) optimizer and learning-rate choices are primary drivers of DD; and (iii) representation-space probes reveal structured geometry that explains why accuracy can recover beyond the interpolation threshold even when loss curves show pronounced peaks. Although the exact numerical characteristics may differ across architectures, the underlying patterns remain consistent, suggesting that the mechanisms driving DD are not peculiar to a specific network family but reflect more general principles of overparameterized neural networks. In fact, we showed that even simpler architectures like FCN networks, which lack the spatial structure of CNNs, exhibit similar trends in how noise amplifies DD and how geometry-based evaluation metrics reveal latent structure. This indicates that insights drawn from one network type could likely be meaningfully extended to others, providing a common foundation for theoretical analyses and practical guidelines.

## D  NEURAL COLLAPSE METRICS

To assess the extent to which Neural Collapse occurs in our setting, we employ several standard metrics proposed by Papyan et al. (2020). The values of these metrics were collected at each training epoch. We provide a brief description of each one below.

**Variation coefficients**   The variation coefficient, also known as relative standard deviation, is a measure of dispersion and it is defined as the ratio between the standard deviation and the mean. In our setting, the coefficient of variation is computed for the globally centered centroids of each class in the latent space of the last hidden layer. Specifically, we calculate the following quantity:

$$\frac{\mathrm{Std}_{c}\left(\|\boldsymbol{\mu}_c\|_2\right)}{\mathrm{Avg}_{c}\left(\|\boldsymbol{\mu}_c\|_2\right)}$$

, where $\boldsymbol{\mu}_c$ is the centroid of the $c$-th class. A convergence of this quantity toward zero indicates that all centroids tend to have the same norm (i.e., they become equinormed), a necessary condition for them to form a simplex equiangular tight frame.

**Standard deviation of cosines**   To assess angular uniformity, we first computed the cosine of the angle between each pair of centroids, and then calculated the standard deviation of these values:

$$\mathrm{Std}_{c,c'\neq c}\left(\cos(\boldsymbol{\mu}_c,\boldsymbol{\mu}_{c'})\right).$$

As training progresses, this metric is expected to decrease toward zero, which would indicate that the angles are becoming uniform.

**Average of shifted cosines**   The maximal separation angle corresponds to all cosine similarities approaching $-\frac{1}{C-1}$, where $C$ is the number of classes. To evaluate this, we computed the average deviation of cosine similarities from this target value for each pair of centroids:

$$\mathrm{Avg}_{c,c'}\left(\left|\cos(\boldsymbol{\mu}_c,\boldsymbol{\mu}_{c'})+\frac{1}{C-1}\right|\right).$$

A value approaching zero reflects convergence toward maximal equiangularity.

**Frobenius norm of weight-centroid difference**   Finally, the alignment between the classifier weights and the class centroids is measured by comparing their normalized matrices. Let $W$ denote the weight matrix and $M$ the matrix whose columns are the class centroids. We computed

$$\left\|\frac{\mathbf{W}}{\|\mathbf{W}\|_F}-\frac{\mathbf{M}}{\|\mathbf{M}\|_F}\right\|_F^2.$$

A decreasing value indicate an increasing proportionality, which is consistent with the final stage of neural collapse, where classifier weights align with centroids.

**Findings**   As previously discussed in Section 4.3, we observe *incomplete* neural collapse across all tested optimizers: metrics decrease during training but never converge to zero (Figs. 13 to 15). Results are reported only for the widest model in each family ($k = 64$). We also verified that smaller widths behave consistently: since neural collapse is not achieved even with the largest model configurations, it does not occur with smaller ones either. The findings confirm that, even in overparameterized regimes where models interpolate the training data, centroids do not collapse fully. Importantly, this incomplete collapse is not a limitation for our study but rather the realistic setting we wish to investigate, where geometry-based accuracies remain informative. These plots further support our interpretation of the observed discrepancies between standard and geometry-based accuracies.

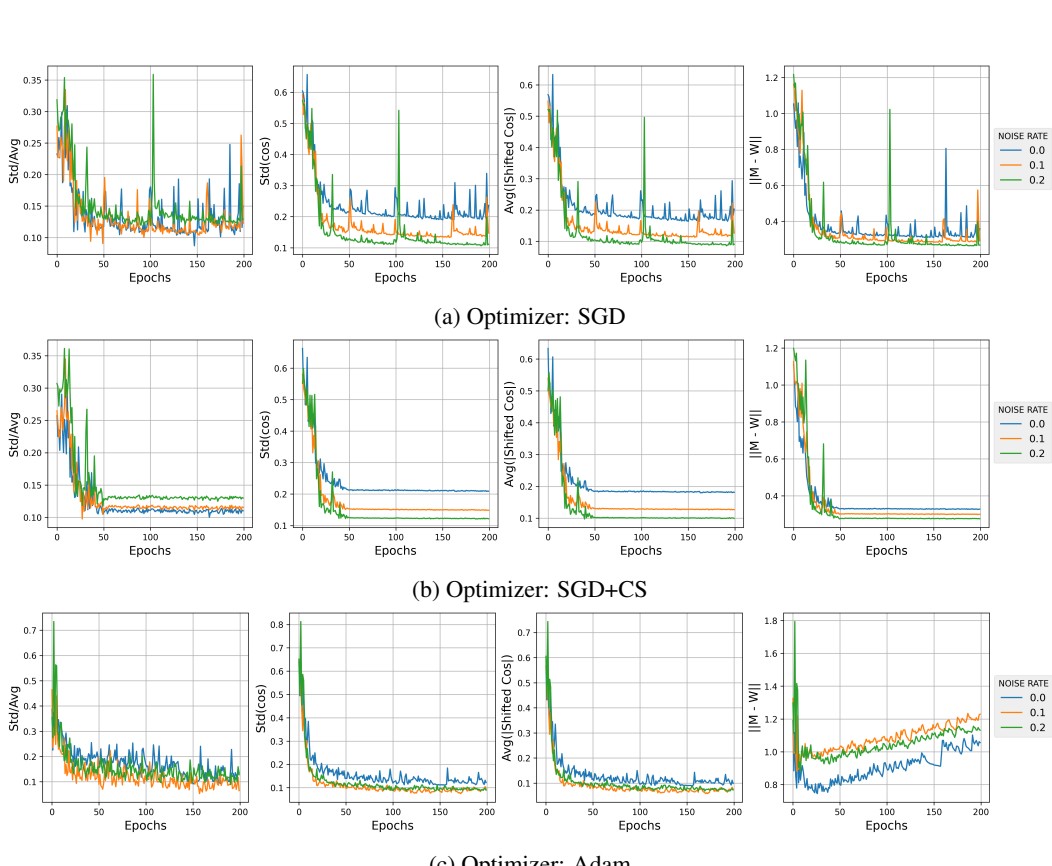

(a) Optimizer: SGD

(b) Optimizer: SGD+CS

(c) Optimizer: Adam

Figure 13: From left to right: (i) coefficient of variation (Std/Avg); (ii) standard deviation of cosine (Std(cos)); (iii) average shifted cosine (Avg(|shifted cos|)); (iv) Frobenius norm of the weight–centroid difference ($\|M - W\|$), all computed on the class centroids. Metrics were obtained by training a ResNet model with base convolutional layer width $k = 64$ on CIFAR-10 using Cross-Entropy loss. Each row in the plot corresponds to a different optimizer, all initialized with a learning rate of 0.01

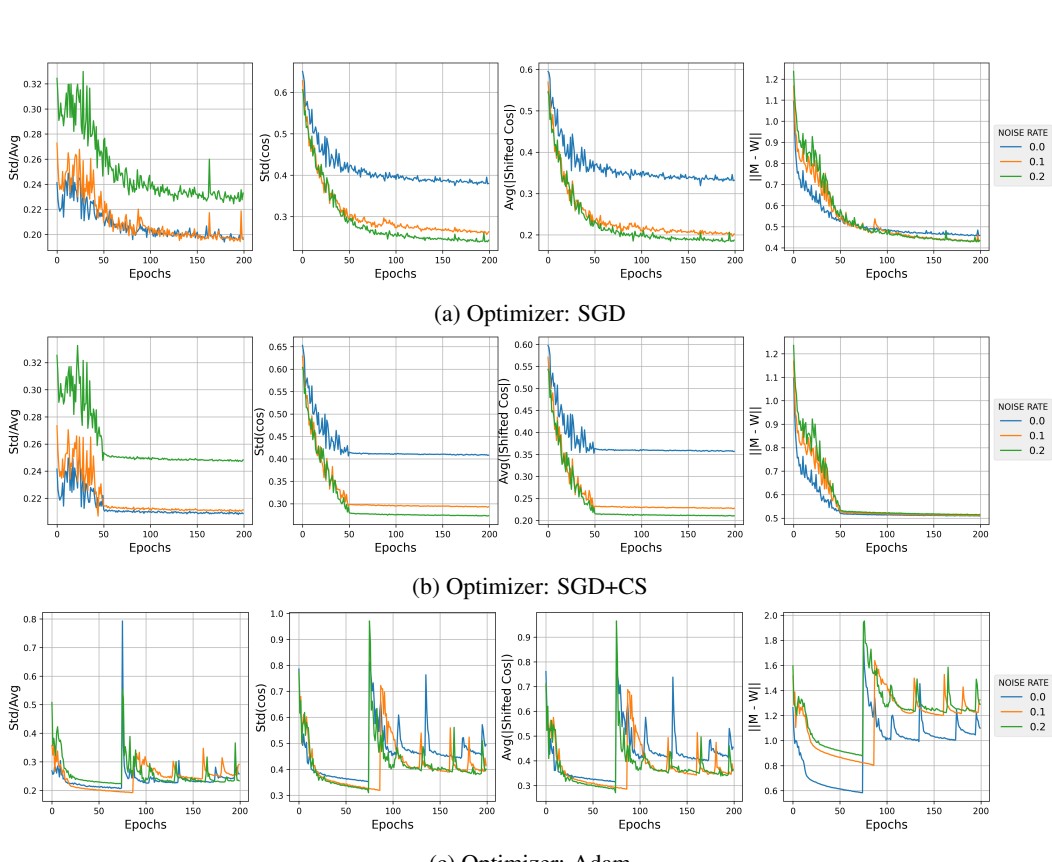

Figure 14: From left to right: (i) coefficient of variation (Std/Avg); (ii) standard deviation of cosine (Std(cos)); (iii) average shifted cosine (Avg(|shifted cos|)); (iv) Frobenius norm of the weight–centroid difference ($\|M - W\|$), all computed on the class centroids. Metrics were obtained by training a CNN model with base convolutional layer width $k = 64$ on CIFAR-10 using Cross-Entropy loss. Each row in the plot corresponds to a different optimizer, all initialized with a learning rate of 0.01

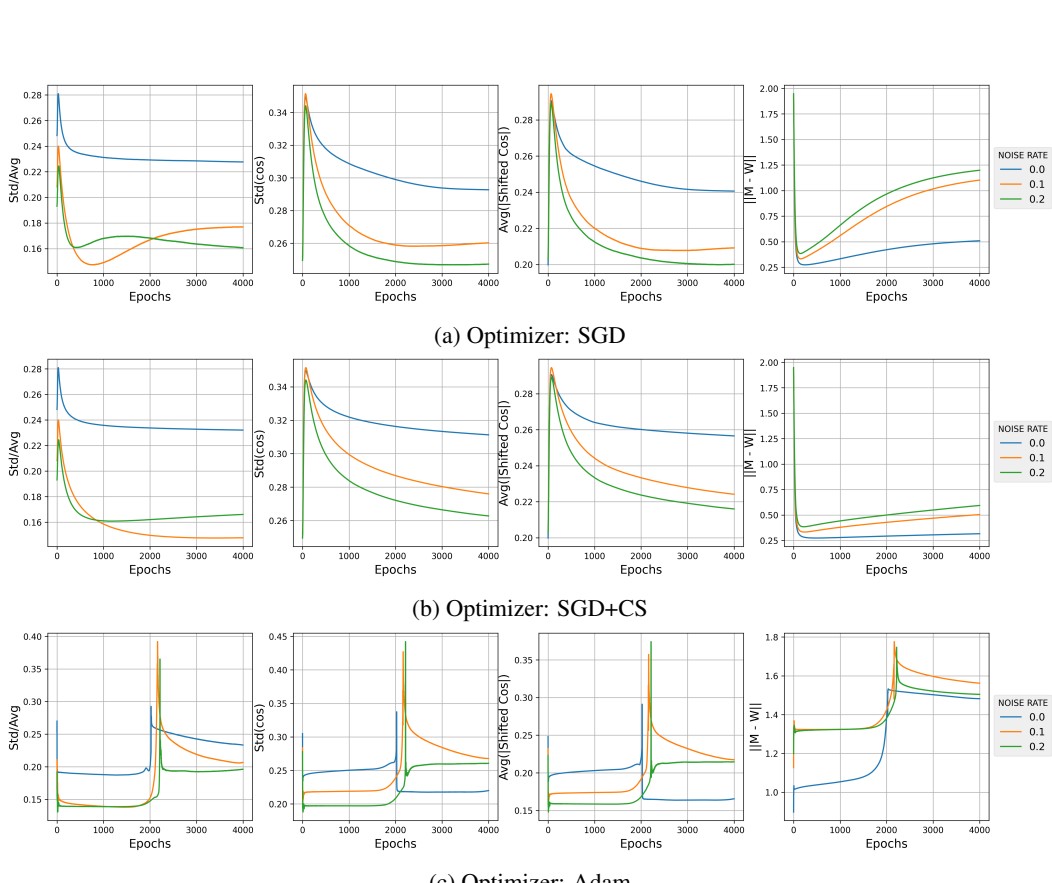

(a) Optimizer: SGD

(b) Optimizer: SGD+CS

(c) Optimizer: Adam

Figure 15: From left to right: (i) coefficient of variation (Std/Avg); (ii) standard deviation of cosine (Std(cos)); (iii) average shifted cosine (Avg(|shifted cos|)); (iv) Frobenius norm of the weight–centroid difference ($\|M - W\|$), all computed on the class centroids. Metrics were obtained by training a FCN model with base hidden layer width $k = 64$ on MNIST using Cross-Entropy loss. Each row in the plot corresponds to a different optimizer, all initialized with a learning rate of 0.01

## E    TRAINING LOSS

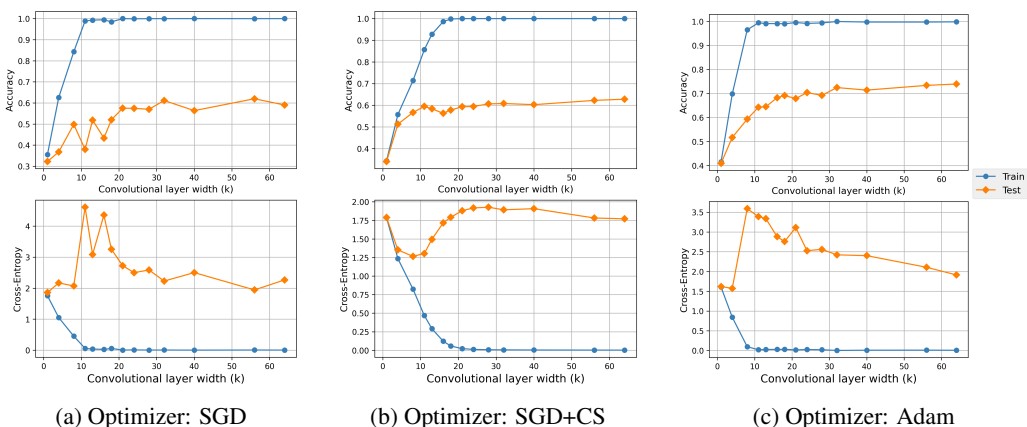

(a) Optimizer: SGD          (b) Optimizer: SGD+CS          (c) Optimizer: Adam

Figure 16: Accuracy (top row) and Cross-Entropy (bottom row) on train and test sets as functions of the base convolutional layer width (k) of ResNet models trained on CIFAR-10 with Cross-Entropy loss. Each row in the plot corresponds to a different optimizer, all initialized with a learning rate of 0.01

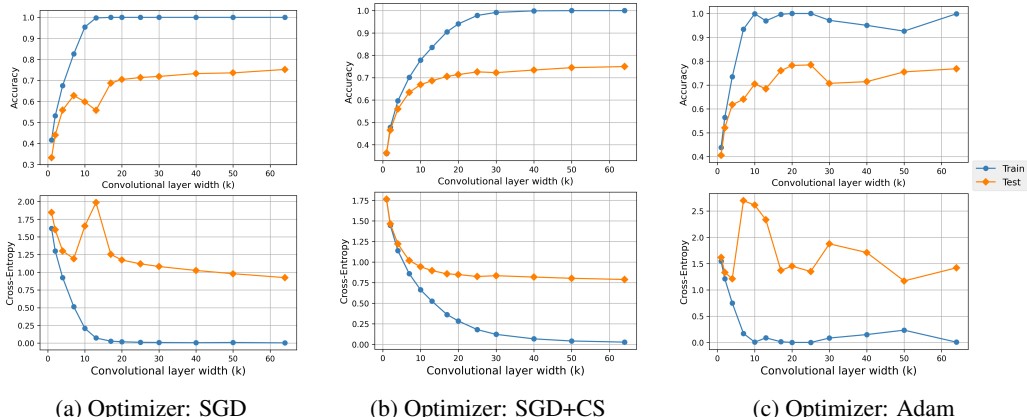

(a) Optimizer: SGD          (b) Optimizer: SGD+CS          (c) Optimizer: Adam

Figure 17: Accuracy (top row) and Cross-Entropy (bottom row) on train and test sets as functions of the base convolutional layer width (k) of CNN models trained on CIFAR-10 with Cross-Entropy loss. Each row in the plot corresponds to a different optimizer, all initialized with a learning rate of 0.01

Figs. 16 to 18 report both training and test quantities: CE and classification accuracy. Concretely, each panel shows these metrics for noise rate 0, training with CE Loss and varying the optimizer. These plots are included as they confirm that our models attain *perfect interpolation* in the overparameterized regime: training CE approaches zero and training accuracy reaches 1.0 for the intermediate- to large-width models used in the paper. This is important because double descent is defined with respect to interpolation behaviour; showing train loss = 0 and train accuracy = 1 documents that the interpolation threshold has been crossed in our experiments.

Figs. 19 and 20 report both training and test quantities for CNN trained on noisy data: CE and classification accuracy. Each panel shows these metrics for noise rates 0.1 and 0.2, training with CE Loss and varying the optimizer. These plots confirm that even if them models attain *perfect interpolation* (training accuracy reaches 1.0) and label noise is present, the emergence of double descent still depends on the choice of optimizer. Specifically, we observe a clear DD peak for Figs. 19a, 19c, 20a and 20c and we whereas no such peak appears for Figs. 19b and 20b.

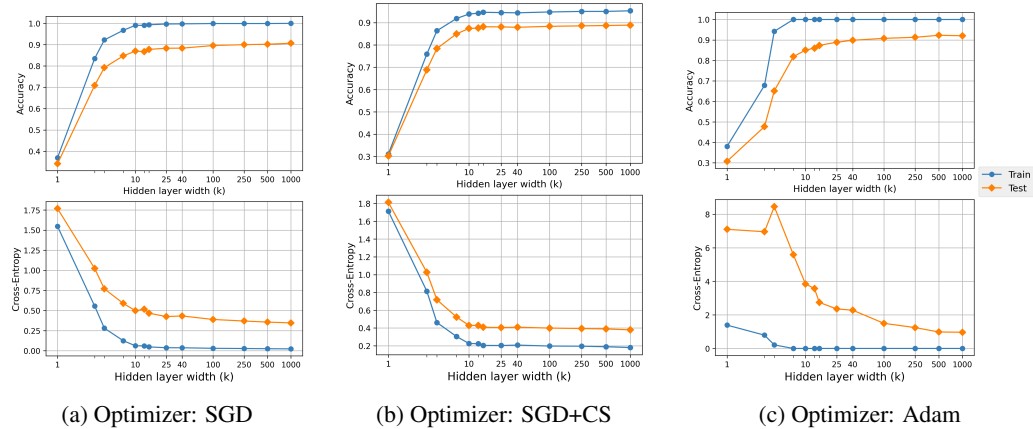

(a) Optimizer: SGD   (b) Optimizer: SGD+CS   (c) Optimizer: Adam

Figure 18: Accuracy (top row) and Cross-Entropy (bottom row) on train and test sets as functions of the base hidden layer layer width (k) of FCN models trained on MNIST with Cross-Entropy loss. Each row in the plot corresponds to a different optimizer, all initialized with a learning rate of 0.01

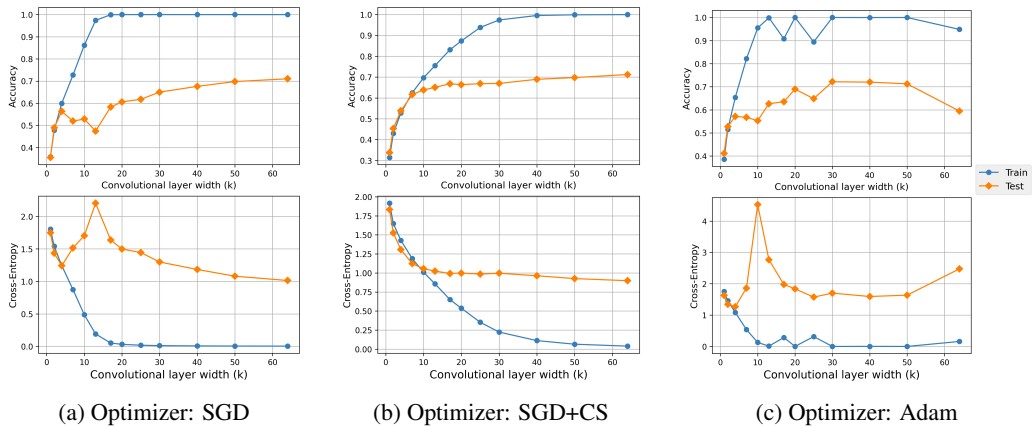

(a) Optimizer: SGD   (b) Optimizer: SGD+CS   (c) Optimizer: Adam

Figure 19: Accuracy (top row) and Cross-Entropy (bottom row) on train and test sets as functions of the base convolutional layer width (k) of CNN models trained on CIFAR-10 (noise rate 0.1) with Cross-Entropy loss. Each row in the plot corresponds to a different optimizer, all initialized with a learning rate of 0.01

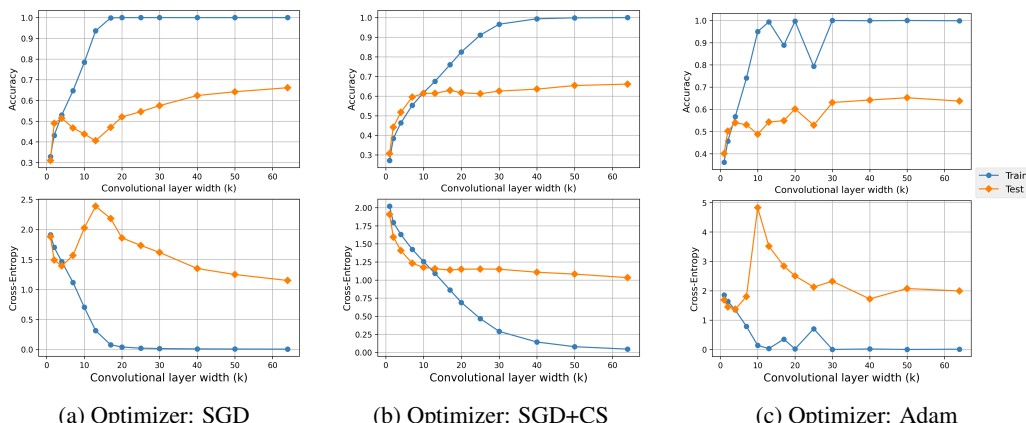

(a) Optimizer: SGD   (b) Optimizer: SGD+CS   (c) Optimizer: Adam

Figure 20: Accuracy (top row) and Cross-Entropy (bottom row) on train and test sets as functions of the base convolutional layer width (k) of CNN models trained on CIFAR-10 (noise rate 0.2) with Cross-Entropy loss. Each row in the plot corresponds to a different optimizer, all initialized with a learning rate of 0.01

## F  MODEL SELECTION (BEST VS LAST EPOCH)

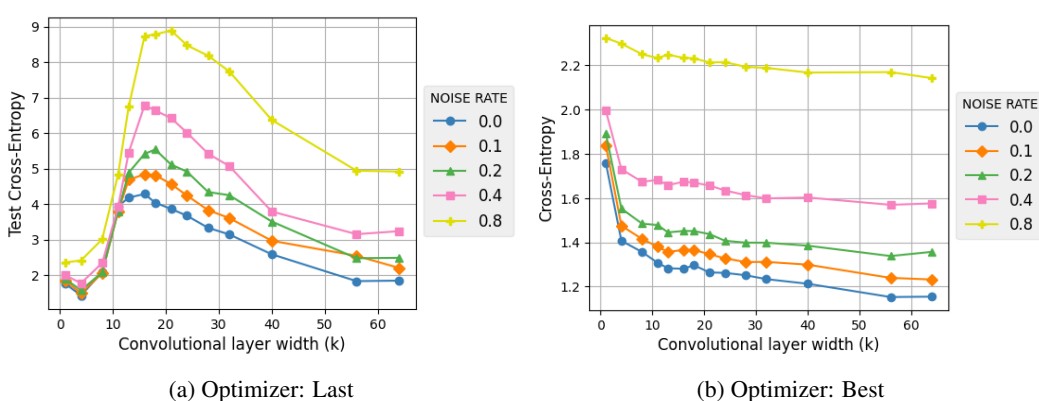

(a) Optimizer: Last

(b) Optimizer: Best

Figure 21: Cross-Entropy on test set as a function of the base convolutional layer width (k) of ResNet models trained with Cross-Entropy loss on CIFAR-10 using Adam with initial learning rate 0.0001. Results compare the Cross-Entropy at the last training epoch (left) versus the best-performing epoch (right).

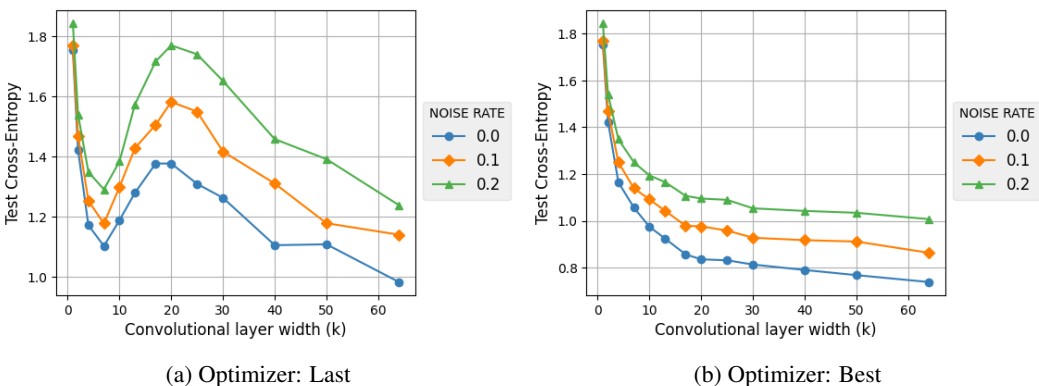

(a) Optimizer: Last

(b) Optimizer: Best

Figure 22: Cross-Entropy on test set as a function of the base convolutional layer width (k) of CNN models trained with Cross-Entropy loss on CIFAR-10 using Adam with initial learning rate 0.0001. Results compare the Cross-Entropy at the last training epoch (left) versus the best-performing epoch (right).

In the main text, we present metrics computed on the final training checkpoint to illustrate the *last-model* behaviour, a common practice in DD studies. To ensure comprehensiveness, this section presents a complementary analysis in which, for each iteration, the checkpoint exhibiting the minimum validation loss (the "best epoch") is selected. This selection protocol is closer to standard practical pipelines, where different techniques (e.g. early stopping) are commonly applied to avoid overfitting, providing a realistic perspective on the phenomenon.

As illustrated in Figs. 21 to 23, the selection of the validation-best checkpoint led to the complete elimination of the DD peak that was observable when evaluating the last-epoch model. This result aligns with the observation made in Section 5 that DD, while theoretically relevant for understanding overparameterized models, may have limited practical impact in typical training settings where validation-based model selection is applied. Due to limitations in time and resources that precluded the execution of a comprehensive experimental protocol, we present these partial results with the hope that they will offer valuable insights to other researchers in the field.

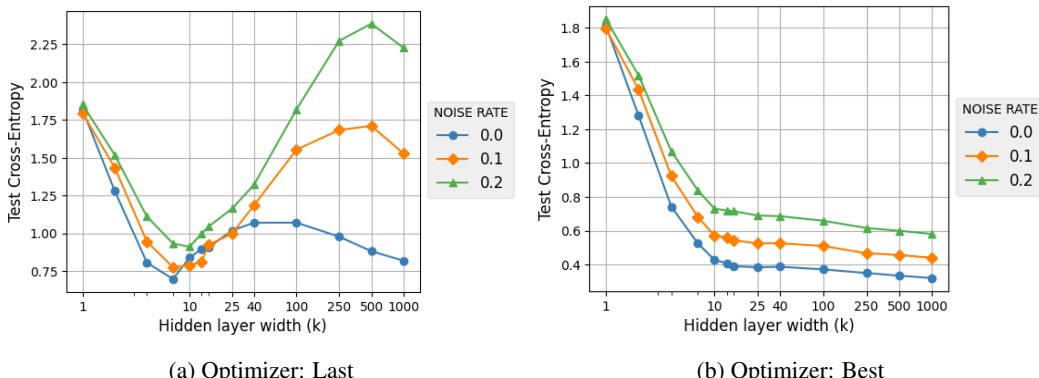

(a) Optimizer: Last          (b) Optimizer: Best

Figure 23: Cross-Entropy on test set as a function of the base hidden layer width (k) of FCN models trained with Cross-Entropy loss on MNIST using Adam with initial learning rate 0.0001. Results compare the Cross-Entropy at the last training epoch (left) versus the best-performing epoch (right).

## G LLM USAGE

Large language models were utilized exclusively for minor support tasks, such as assisting with code debugging and enhancing the quality of written content. All core aspects of the research were conducted entirely by the authors. LLMs were not used to generate, reinterpret, or alter any substantive scientific content.

