# OpenReview forum: "Double Descent Revisited: When Noise Amplifies and Optimizers Decide"
_ICLR.cc/2026/Conference — Submitted to ICLR 2026_

### Official Review · Reviewer_Vzyp · 2025-10-23

**Soundness:** 1
**Presentation:** 2
**Contribution:** 2
**Rating:** 4
**Confidence:** 3

**Summary:**

The paper investigates the underlying causes of the double descent phenomenon in deep learning. The paper challenges the widely held belief that label noise is the primary driver of the interpolation peak observed in the double descent curve. Through a series of experiments, the paper demonstrates that optimization dynamics, particularly the choice of optimizer and learning rate schedule, play a more significant role in the emergence of double descent than the presence of label noise. Furthermore, the paper analyzes the geometry of learned representations and finds that even when the final classifier struggles, the model can learn robust representations.

**Strengths:**

The paper identifies that the optimizer and learning rate also play an important role in double descent, not solely the noisy label.

**Weaknesses:**

Despite the claim that optimizers decide the double descent is promising, the following concerns must be adequately addressed for the paper to be accepted.

1. The paper's logic is not rigorous enough. From the title "Optimizer decides" and line 65 "On the contrary, the optimization algorithm and
learning rate (with and without scheduler) appear to be strong drivers of DD" in introduction, it seems to argue that the optimizer is the dominant factor in double descent. However, the experimental results may not fully support this claim. In Section 4.1, while figures (a) and (b) lack a distinct peak, a clear plateau is observed at around the 10th layer. This might only indicate a less obvious double descent, not its absence. Furthermore, Figure 2 in [1] demonstrated that SGD can also produce a clear double descent phenomenon, so the results here cannot conclusively position the optimizer as the core driver. It is possible that different optimizers, similar to label noise, have varying strengths of influence. I think at best, Section 4.1 demonstrates that the optimizer plays a role **but not the key role** in double descent. The authors should clarify this point.
2. If the above point 1 is valid, meaning both the optimizer and label noise are drivers, then the novelty of this paper becomes questionable, as its experimental setup and the source codes are nearly identical to that of [1].
3. To convincingly prove that the optimizers "decide" the phenomenon, relying solely on experimental evidence is insufficient, as exceptions can always exist. A better approach would be to incorporate a theoretical component, demonstrating that under certain settings, the use of an optimizer with specific properties theoretically guarantees the emergence of the double descent phenomenon, even in the absence of label noise.

Reference.

[1] Gu, Y., Zheng, X., & Aste, T. (2023). Unraveling the Enigma of Double Descent: An In-depth Analysis through the Lens of Learned Feature Space. ArXiv, abs/2310.13572.

**Questions:**

See above.

---

> ### Author Response · Authors · 2025-11-19
>
> W1:We do argue that optimizer related choices (optimizer, scheduler, and learning rate) appear to be the main drivers in double descent, at least when compared to noise.
> As to the remark that "while figures (a) and (b) lack a distinct peak, a clear plateau is observed at around the 10th layer. This might only indicate a less obvious double descent, not its absence": this seems unlikely to us for a number of reasons.
> First, the maximum range of width values we consider corresponds to a number of parameters that is well beyond the number of points (the interpolation threshold is around 10 in our case, where the number of trainable parameters reaches 41940, against 40000 training samples) where the peak of double descent is typically observed.
> Second, Fig. 2 shows that a DD peak does appear for test accuracy in this same width range, suggesting that the chosen scale is appropriate. Third, we adopt exactly the same width range used in [1], where a DD peak also appears around width ≈10.
> Our results do not contradict the results in [1]; rather, they indicate that noise alone may not reliably produce DD, and that optimization choices can strongly modulate whether DD appears. Indeed our results show systematic cases where DD appears without noise (Fig. 1c, Fig 16, Fig 17a,c , Fig 18c) and cases where DD fails to appear despite significant noise (Fig. 1a–b, Fig 19 b, Fig 20b) which challenges the idea that noise plays the primary role.
>
> W2: Our work is complementary to [1]. While [1] provides a geometric interpretation that focuses on the effect of noise, our study systematically examines  when DD appears or disappears as one varies optimizer, scheduler, and learning rate, dimensions that [1] does not investigate in depth. We believe this broader exploration of optimization-related factors meaningfully extends the empirical picture. Please, see also our answer to Q2 of reviewer  VWWQ.
>
>
> W3:  We acknowledge this limitation.  At present, however, existing theoretical analyses are still limited and typically focus on simpler models rather than deep architectures. At the same time, many influential studies in this area have begun by establishing careful empirical evidence before a complete theory emerges and our work falls in this line of study, like many others in the field.
> We view our work as part of a broader empirical effort, similar to many influential studies on DD, including [1], that aims to provide careful observations that can guide future theoretical developments. We agree that theoretical progress is essential; the main goal of our study is precisely to offer concrete observations that could meaningfully guide future theoretical investigations.

---

> > ### Comment · Reviewer_Vzyp · 2025-11-23
> >
> > Dear Authors,
> >
> > Thank you for your response. I acknowledge the significant experimental effort you have invested in investigating Double Descent, and I agree with your perspective that label noise is not the dominant factor. However, while your work successfully identifies correlational relationships between optimization choices (like the optimizer and learning rate) and the phenomenon, the study remains at this empirical level. I believe the research could be strengthened by moving towards a more causal and quantitative understanding. For instance, constructing a simplified, idealized setting to develop a theoretical framework that can quantify the impact of these individual factors on Double Descent would provide deeper insight. Therefore, I tend to maintain my original score.

---

### Official Review · Reviewer_VWWQ · 2025-10-27

**Soundness:** 3
**Presentation:** 3
**Contribution:** 2
**Rating:** 2
**Confidence:** 3

**Summary:**

This paper presents an empirical investigation of model-wise double descent across a wide range of settings. The authors conduct ablation studies on key factors, including noise level, loss function, optimizer, and learning rate, to understand the respective roles of noise and the optimization process.

Two primary findings emerge from their analysis. First, the double descent phenomenon is observed even in the absence of label noise (i.e., with clean data). Second, the choice of optimizer and learning rate significantly alters the double descent curve, sometimes causing the effect to vanish entirely. Lastly, they also check the properties of the representations and provide links to neural collapse phenomena.

**Strengths:**

The paper's extensive experimentation allows for a detailed discussion of the various factors contributing to double descent. The work is clearly written and accessible.

**Weaknesses:**

1. The analysis seems to be in line with previous work, and its contribution beyond the existing literature is unclear. The authors write, "Only a handful of studies have emphasized the role of optimization dynamics," but they provide numerous previous works that consider this perspective. Moreover, there is a huge literature on implicit bias in overparameterized networks which also suggests an explanation for DD via optimization dynamics.

2. The analysis assumes the datasets used are noise-free. I am not sure if this is true (at least for CIFAR-10), see [1]. Moreover, it is unclear to me what the added benefit is of experiments that show double descent when there is no explicit label noise. These findings are also present in [2].

[1] Wei, Jiaheng, et al. "Learning with noisy labels revisited: A study using real-world human annotations." arXiv preprint arXiv:2110.12088 (2021).
[2] Gu, Yufei, Xiaoqing Zheng, and Tomaso Aste. "Unraveling the enigma of double descent: An in-depth analysis through the lens of learned feature space." arXiv preprint arXiv:2310.13572 (2023).

**Questions:**

1. Can you clarify the statement: "DD is not caused by noisy data but ... is directly attributable to the optimization process"? Doesn't the noise level directly alter the optimization process? I would understand the following claim better: "DD is still present in non-noisy data".

2. Can you explain the added contribution over [2]? I think their statement matches the statement presented in this paper which is that noise amplifies DD. [2] also argues that there is label-noise in MNIST and CIFAR-10. Did you take into account any label-noise that was already present in the dataset labeling in your analysis?

---

> ### Author Response · Authors · 2025-11-19
>
> W1: While several prior works have touched on the role of optimization dynamics, to the best of our knowledge there has been no systematic and comprehensive study of this aspect in the context of double descent. Our work investigates a substantially broader range of factors than is typically considered, including three different optimizers (each with multiple learning rates), multiple loss functions, several levels of label noise, and three distinct architectures. This breadth allows us to probe the phenomenon more thoroughly and to reveal interactions that are difficult to appreciate from narrower studies.Regarding implicit bias, we agree that the literature is extensive and that it offers promising theoretical tools. Regarding implicit bias, we agree that the literature is extensive and that it offers promising theoretical tools. However, despite this progress, it remains unclear how implicit bias theories can fully account for empirical behaviors such as those presented in Section 4 of our study (and in earlier empirical work). In fact, the very nature of double descent is still debated: noise is often highlighted as a key factor, but alternative hypotheses, and combinations of factors, remain viable in many settings. For example, as we note in Section 5, recent results (e.g., Curth et al., NeurIPS 2023) show that certain instances of double descent can indeed be framed within the classical bias–variance tradeoff for simpler models. Nevertheless, these explanations do not yet extend robustly to the more complex scenarios we study. Our primary goal was therefore to build on the contribution of [2], emphasizing that the landscape is more nuanced than previously assumed and that the optimization process itself may play a central role in the emergence of double descent.
>
> W2:  We appreciate the reviewer’s insightful comments. Indeed, it is true that datasets like CIFAR-10 are not entirely noise-free. However, the estimated level of inherent noise is relatively modest and significantly lower than the injected noise levels we study. Our objective is specifically to investigate the impact of label noise, so including experiments with both 0% injected noise and higher noise levels is appropriate and follows common practice in this line of research. Furthermore, while double descent has been observed in noise-free settings in prior work [2], we believe our results provide new insights. In particular, we show that the presence or absence of double descent can depend not only on the amount of noise but also on the choice of the optimizer. For example, we observe that under the same noise conditions, some optimizers exhibit double descent while others do not, and remarkably, this holds even in the absence of injected noise. We find this behavior both surprising and noteworthy, and believe it represents an important contribution of our work.
>
> ---
>
> Q1: Our statement is not meant to deny that label noise can affect optimization dynamics, indeed, it clearly does. Rather, our point is that label noise does not seem to be the primary or sufficient cause of double descent (DD) in our experiments. Specifically, (i) DD consistently appears even when no label noise is present at all (Fig. 1c, Fig 16, Fig 17a and 17c , Fig 18c), and (ii) in several other settings, DD does not appear despite the presence of substantial label noise (Fig. 1a–b, Fig 19 b, Fig 20b).  These two observations suggest that label noise alone cannot explain the emergence of DD, and that its role may be secondary or amplifying rather than foundational.
>
> Q2: We fully agree with [2] and prior work that label noise tends to amplify double descent, and we do not claim any novelty for that observation. Our contribution is different in focus. While [2] provides a compelling geometric interpretation, showing that overparameterized models interpolate noisy samples by placing them in isolated regions of feature space, their analysis does not investigate when this behaviour appears or under what conditions it can disappear. Our work is complementary: we systematically examine how optimization-related factors (in particular, the choice of optimizer, scheduler, and learning rate) influence the emergence or suppression of DD across multiple architectures. Our results show that the same model trained on the same dataset can either exhibit or fail to exhibit DD purely as a consequence of these optimization choices. Regarding inherent dataset label noise, we note that [2] argues that datasets such as MNIST and CIFAR-10 contain non-negligible accidental label noise. This is plausible, but such inherent noise is naturally included in our experiments: we do not artificially “clean” these datasets.
> At a minimum, our findings suggest that optimization-related factors play a crucial role in determining when the geometric behaviour described in [2] emerges. This line of inquiry does not appear in [2], and we view our work as expanding rather than contradicting that perspective.

---

> > ### Comment · Reviewer_VWWQ · 2025-11-26
> >
> > I am willing to increase my score to 4 if the authors polish the phrasing on label noise - optimization landscape (as they have done in the rebuttal) and b) provide the necessary evidence regarding the low inherent label noise claim. I believe these are necessary steps for the integrity and clarity of the paper.

---

### Official Review · Reviewer_4ydv · 2025-10-28

**Soundness:** 4
**Presentation:** 3
**Contribution:** 2
**Rating:** 6
**Confidence:** 5

**Summary:**

This paper revisited the double descent phenomenon of deep neural networks, and analyzed the impact of the optimizer and noise. With verifying and rejecting prior hypotheses, the study confirmed that the choice of optimizer decides the phenomenon occurence and noisy regimes amplifies the phenomenon.

**Strengths:**

1. This paper revisited the optimization perspective in the DD phenomenon and evaluated from both loss functions / noise factors and geometry -based metrics, demonstrated in-depth understanding and analytics towards the problem.
2. The experiment design is complete and carefully implemented , with code open-sourced for re-production.

**Weaknesses:**

1. While the paper discussed the different role of optimizer and noise w.r.t. the DD phenomenon, it does not discussed how different optimizer handle noisy regimes from a fundamental perspective (only gently touched in the 4.3 section), which degrades the depth of analytics.
2. As DD is first reported and mostly observed in deep learning and DNNs, the role of machine learning model and task characteristics is also critical to the analytics of DD. I believe the conclusion of "Optimizers Decide" is too vague in a bigger landscape.

**Questions:**

1. As discussed in the weaknesses, can you hypothesize how different optimizers promote learning in a noisy regime, which decides the DD phenomenon, according to your hypothesis?
2. As a question, is it possible to extend your analytics to more recent large language models? I believe frameworks like NanoGPT are relatively accessible in comparison to CNNs, and either verifying the DD phenomenon or your hypothesis would be an extension to your existing works.
3. While a unified theory remains absent towards the DD phenomenon, suggestions on future works and the remaining questions from this paper's perspective may also be a valuable contribution to the field.

---

> ### Author Response · Authors · 2025-11-19
>
> W1: We thank the reviewer for this comment. It would be nice to have a comprehensive and compelling picture of double descent, even though achieved only through experiments. As we embarked in this study, our first goal was to investigate the impact of noise robust losses on double descent mitigation, under the conventional wisdom that noise is the key factor behind double descent. We soon discovered that the picture looks much more nuanced than it appears and that seemingly minor changes to the optimization algorithm (such as a slight increase or decrease in learning rate) can induce or completely remove the phenomenon, all other being equal. We dug deeper and, within the limits of available resources, we explored several dimensions, including noise of course. The picture that emerges does not seem trivial to decipher, but it points in a general direction, i.e., the optimization dynamics, something we emphasize throughout our paper. We believe that achieving a deeper, more fundamental theoretical understanding will require time and further research. Our goal here was mainly to highlight some phenomena and discrepancies wrt conventional wisdom that in our opinion might guide further theoretical research on the topic.
>
>
> W2: We totally agree that model architecture, task complexity, and dataset characteristics play a role in shaping double descent, and we do not intend to downplay their importance. Our goal was instead to highlight that, even when these factors are held fixed, the optimization process alone can determine whether a visible DD peak emerges. Across all architectures we tested (FCN, CNN, ResNet) and across both MNIST and CIFAR-10, we consistently observed that the presence or absence of DD seems to depend far more strongly on optimizer choice and learning-rate dynamics than on the model family itself. For example, Adam exhibits a clear DD peak for every architecture and dataset, even in (injected) noise-free settings, whereas SGD with small learning rates does not exhibit DD even under extreme label noise.
>
> ----------------------------
>
> Q1: Please see answer to W1 above.
>
>
> Q2: We fully agree that language models, such as GPT-based architectures, represent a highly relevant and central direction of research in our days. However, how to explore DD phenomenon for LLMs is still an under-investigated problem and a potentially very interesting direction of research; studying it systematically in LLMs remains extremely challenging due to the computational cost of training multiple variants from scratch and the lack of publicly available intermediate checkpoints.
>
>
> Q3: This is what we attempted in Section 5 (“Discussion, Limitations, and Conclusions”), where we discussed some open questions and potentially interesting directions for future research. Among others, we would like to mention the disappearance of double descent when best model selection is applied, something that could suggest a different angle from which to investigate the phenomenon, both empirically and theoretical; the connection between double descent and neural collapse, which might drive a more in-depth study in feature space. More broadly, we hope that our extensive effort to connect existing observations and reconcile different hypotheses will help and inspire further research toward a comprehensive theory of this phenomenon.

---

### Official Review · Reviewer_URSt · 2025-10-30

**Soundness:** 2
**Presentation:** 3
**Contribution:** 1
**Rating:** 2
**Confidence:** 5

**Summary:**

The paper discusses the double descent phenomenon, which is the observation that test error can increase near the interpolation threshold and then decrease again as model capacity increases. While past work attributed DD primarily to label noise, this study argues that optimization dynamics (optimizer type, learning rate, scheduler) are the real reasons, with noise acting mainly as an amplifier. The authors also show that noise primarily affects the linear separability of different classes in feature space.

**Strengths:**

Strengths:

1. The authors challenge the prevailing idea that label noise causes double descent, showing instead that optimization dynamics are the main reason for double descent phenomena.

2. The paper performs a well-controlled empirical sweep across optimizers, learning rates, and schedulers, isolating their effect on double descent.

3. Introducing geometry-based metrics (k-Nearest Neighbour and Nearest Centroid accuracies) provides a deeper probe into how learned features evolve across parameter regimes.

**Weaknesses:**

Weaknesses:

1. The paper lacks a theoretical framework explaining why optimizer dynamics cause double descent, and how noise is not a necessary condition for double descent phenomena.

2. The analysis focuses mostly on learning rate and optimizer type, but other factors (batch size, momentum, weight decay) are not explored.

3. Experiments use only MNIST and CIFAR-10, which may not capture behaviors of larger or more realistic models.

4. The paper focuses almost exclusively on test-set curves (test loss and test accuracy) when illustrating the double descent phenomenon. However, by definition, double descent describes a mismatch between training and test behaviors. Specifically, the training loss continuously decreases or saturates at near-zero, while the test loss first decreases, then increases near the interpolation threshold, and decreases again afterward. Without showing training loss/accuracy, it is not possible to confirm that the observed “double descent” peaks are truly due to the generalization gap rather than optimization instability or noise.

5. The paper lacks quantitative metrics and statistical analysis measures

**Questions:**

Questions to the Authors:

1. Could you provide training loss and accuracy curves to confirm that the models actually interpolate (zero training loss) near the test loss peak?

2. Statistical Significance: Did you average the results over multiple random seeds?

3. Authors should discuss the theoretical reasons behind why noise is not a necessary condition for the double descent phenomenon. Why do optimizer dynamics cause double descent?

4. For ResNet, how does the double descent behavior change if you scale depth (number of blocks) rather than width?

---

> ### Author Response · Authors · 2025-11-19
>
> W1:  We agree that a deeper theoretical explanation of why optimizer dynamics can induce double descent would significantly strengthen the overall understanding of the phenomenon.
> We acknowledge this limitation.
> However, existing theoretical analyses (see also our related work section) are still limited and typically focus on simpler models rather than deep architectures. At the same time, many influential studies in this area have begun by establishing careful empirical evidence before a complete theory emerges and our work falls in this line of study, like many others in the field. In particular, it aims precisely to offer concrete observations that could meaningfully guide future theoretical investigations.
>
>
> W2: We are aware of these and other limits of this study, which we discuss in Section 5 with a level of detail compatible with page limits.
> We would like to emphasize that our study considers already more “dimensions” (factors that can influence the presence of DD) than typically considered in most previous works, this includes: three different optimizers (with six different learning rates), multiple loss functions, multiple noise rates, and three different architectures.
> Increasing the number of factors one considers has, beyond potential benefits, a couple of serious drawbacks: 1) the complexity of the experimental setup grows exponentially with the number of potential factors one considers; 2) related to this, it becomes increasingly difficult to discriminate the impact of different factors, so that complex and lengthy ablation studies become necessary at the end. We preferred to focus on optimization related factors (optimizer, scheduler and learning rate), which have been mentioned as playing a potential role in DD in previous literature.
>
>
>
> W3: Our choice of datasets follows the established experimental protocol used in the main prior works on model-wise double descent, including Belkin et al. (2019), Nakkiran et al. (2021a), and Gu et al. (2024), which all study the phenomenon primarily on these benchmarks. We believe that using the same datasets allows for a clean comparison.
>
>
> W4: We totally understand the reviewer's concerns. Precisely for this reason in our original submission we had included training and test loss and accuracy in Appendix E.
> We have now expanded the section by adding plots number 19 and 20 in the appendix E. As shown in all these plots, the model interpolates (the training accuracy reaches 100%); however in some cases we have DD in others not depending on the optimizer and despite the presence of injected noise.
>  As you can imagine, exploring every combination of model, width, loss function, optimizer, and learning rate required running a very large number of experiments. Including the plots for each individual run would have taken up an impractical amount of space in the paper.
>
>
> W5: We acknowledge this limitation. However, due to the extensive number of experiments conducted, and considering that many related papers also do not perform statistical tests, we opted instead to demonstrate consistency in model behavior by running multiple experiments across a range of configurations. This approach allowed us to validate the robustness of our findings without relying solely on traditional statistical metrics.
>
> -----------------------------------------
>
> Q1: See also answer to W4.
>
>
> Q2: For the same reasons mentioned above, we were unable to perform multiple runs for every possible combination, since it would have been computationally infeasible. In particular, the number of different configurations we considered is in the order of 1000, which required weeks. Repeating each experiment even 5 times would have been unfeasible.
>
>
> Q3: We agree that a deeper theoretical explanation of why optimizer dynamics can induce double descent would significantly strengthen the overall understanding of the phenomenon. Our main goal here was more humbly to highlight results that to some extent challenge conventional wisdom on double descent, with the hope that this can help guide future theoretical efforts.
>
>
> Q4: The common practice in previous work on the double descent phenomenon is to vary the width of the models rather than their depth. Furthermore, given the number of experiments conducted, exploring both dimensions (width and depth) simultaneously would have involved computational costs that were too high in relation to the available resources. We therefore chose to focus on a broader set of different models in order to verify whether the patterns observed in one would also recur in the others. Finally, it is not quite clear what adding a further dimension (depth) would really add, given the focus of the paper, which is to investigate whether double descent can be explained only in terms of a model’s response to noise or are other, possibly more important, factors related to optimization are involved.

---

> ### Comment · Reviewer_URSt · 2025-11-27
> **Rebuttal Response**
>
> I thank the authors for their detailed response. The authors have addressed my queries; however, I agree with reviewer Vzyp that this study is primarily empirical and would benefit from a clear theoretical understanding of how noise affects learning dynamics. In view of the author's response and other reviews, I maintain my score.

---

### Author Response · Authors · 2025-12-03
**Rebuttal Summary for the Area Chair**

Dear AC and Reviewers,

We thank all the reviewers for their feedback. Below we summarize the main concerns raised and how we addressed them in our rebuttal.

**Reviewer URSt** appreciated the controlled experimental design and the geometry-based analysis, but raised concerns regarding the lack of theoretical grounding, limited experimental factors (e.g., batch size, momentum), dataset scope, the need to show training curves to confirm interpolation, statistical significance.
We acknowledged the absence of a full theoretical framework, positioning our work within the tradition of DD empirical studies.
DD research typically follows one of two complementary paths: practical exploration through empirical studies or rigorous theoretical analysis (that typically focus on simpler models rather than deep architectures).
Our experimental sweep already spans more dimensions than most prior DD studies  (optimizers × learning rates × losses × noise × architectures).
The reviewer also raises concerns about the fact that we run experiments only on MNIST and CIFAR-10 . We justified the dataset choices by aligning with influential and recent works on DD, including Belkin et al. (2019), Nakkiran et al. (2021a), and Gu et al. (2024), for a clean comparison.
We pointed out that training curves were already included in Appendix E and expanded this section with additional plots confirming interpolation across settings.
Regarding the statistical significance of our results and the lack of multiple seeds, we explained the computational infeasibility of repeating ~1000 configurations.

**Reviewer 4ydv** praised the depth of analysis, completeness of the experimental setup, and open-sourcing of code, but asked for deeper discussion of why different optimizers behave differently in noisy regimes, the broader role of model/task characteristics, possible extension to LLMs and suggestions for future work.
We clarified that our primary goal was to highlight some phenomena and discrepancies wrt conventional wisdom while acknowledging that a full theoretical explanation remains open. We emphasized that architecture and task complexity indeed matter, but our results show that optimization choices alone can toggle the presence or absence of DD even when these are fixed. Regarding LLMs, we noted that systematic DD analysis in modern LLMs remains extremely challenging, mainly due to computational costs and lack of intermediate checkpoints.

**Reviewer VWWQ** commended the breadth of experiments, but questioned the incremental contribution relative to prior work, particularly regarding optimization’s role, and highlighted the presence of intrinsic label noise in considered datasets. In response, we clarified that while prior works touch on optimization (especially on the theoretical side), none provide a systematic, multi-dimensional sweep over optimizers, schedulers, learning rates, noise levels, losses, and architectures. Concerning dataset noise, we acknowledged that CIFAR-10 is not perfectly clean but explained that our goal is to study externally controlled noise, following standard DD methodology. Importantly, we highlighted our empirical finding whereby DD can arise without any injected noise and can fail to arise even under substantial noise (such as 80%!), supporting our claim that noise is not the primary driver.

**Reviewer Vzyp** recognized the promise of our claim that optimization choices play a central role in shaping DD; however they questioned whether our evidence is strong enough to support the statement that optimizers “decide” the phenomenon.
Moreover they asked if some of our curves might reflect a weak DD rather than its absence, and expressed concerns regarding novelty relative to Gu et al. (2024) as well as the lack of theoretical guarantees.
We explained why the plateau observed in the reviewer’s example is highly unlikely to reflect a hidden DD peak, given that: (i) our width range extends well beyond the interpolation threshold; (ii) a DD peak does emerge for accuracy in the same width range; and (iii) the scale matches that used in Gu et al. (2024), where a peak is visible.
Regarding novelty relative to Gu et al. (2024): while Gu et al. focus on how noisy samples shape DD through feature space analysis, we investigate fundamentally different questions. We systematically compare multiple noise-robust loss functions and three optimizers (analyses completely absent from Gu et al.) revealing how label noise and optimization algorithms jointly shape DD.
Regarding theoretical guarantees the answer was similar to the one given to reviewer URSt.

In summary, we believe the reviewers’ feedback has helped sharpen the contribution and clarify our positioning within the DD literature.

We thank the Area Chair and the reviewers for their time and constructive comments, and we hope that our clarifications and additional analyses assist in the final evaluation of our work.

The Authors

---

### Meta-Review · Area_Chair_V4hi · 2025-12-21

**Summary:**

Reviewers were concerned that the main claim was too strong relative to the evidence provided. The work is mostly empirical without a theoretical or causal account of why optimization choices induce double descent. Reviewers were also worried about the novelty over prior work and pointed out limited scope (only MNIST and CIFAR-10). The ratings are 2624, tending towards rejection, and I predict that many reviewers would have maintained their score if we have a full discussion period.

**Reviewer Concerns:**

After the rebuttal period, reviewers continued to express concerns that the paper lacks a clear theoretical or causal mechanism explaining why optimization choices induce or suppress double descent, and that the empirical evidence remains limited in scope. One reviewer suggested that extending the study to language models might be feasible with modest compute by using NanoGPT (especially since prior work (e.g., Nakkiran et al., 2021a) already included transformer experiments.)

**Reviewer Scores:**

I believe many would have maintained their score, given the comments above.

---

### Decision · Program_Chairs · 2026-01-26

Reject